EMBO
Molecular Medicine

# A novel model of glioblastoma recurrence to identify therapeutic vulnerabilities

Sara Lucchini [1,5], James G Nicholson [1,5], Xinyu Zhang[1], Jacob Househam [2], Yau Mun Lim[3], Maximilian Mossner[2], Thomas O Millner[1,4], Sebastian Brandner [3], Trevor Graham[2] & Silvia Marino [1,4 ✉]

## Abstract

**Glioblastoma remains incurable and recurs in all patients. Here we design and characterize a novel induced-recurrence model in which mice xenografted with primary patient-derived glioma initiating/ stem cells (GIC) are treated with a therapeutic regimen closely recapitulating patient standard of care, followed by monitoring until tumours recur (induced recurrence patient-derived xenografts, IR-PDX). By tracking in vivo tumour growth, we confirm the patient specificity and initial efficacy of treatment prior to recurrence. Availability of longitudinally matched pairs of primary and recurrent GIC enabled patient-specific evaluation of the fidelity with which the model recapitulated phenotypes associated with the true recurrence. Through comprehensive multi-omic analyses, we show that the IR-PDX model recapitulates aspects of genomic, epigenetic, and transcriptional state heterogeneity upon recurrence in a patient-specific manner. The accuracy of the IR-PDX enabled both novel biological insights, including the positive association between glioblastoma recurrence and levels of ciliated neural stem cell-like tumour cells, and the identification of druggable patient-specific therapeutic vulnerabilities. This proof-of-concept study opens the possibility for prospective precision medicine approaches to identify target-drug candidates for treatment at glioblastoma recurrence.**

**Keywords** Glioblastoma; Recurrence; Mouse Model; Precision-medicine; Cilia
**Subject Categories** Cancer; Chromatin, Transcription & Genomics; Neuroscience

See also: H Miletic & T Daubon

## Introduction

Glioblastoma, IDH-wildtype is the most common primary brain tumour in adults (Louis et al, 2016; Miller et al, 2021; Ostrom et al, 2016). Current standard of care includes maximal safe surgical resection followed by adjuvant radiotherapy (RT) and chemotherapy with temozolomide (TMZ), an alkylating agent that causes DNA-damage via the transfer of methyl groups to DNA bases. Despite this extensive multi-modal clinical intervention, the average survival for glioblastoma patients is only 15 months, and patients invariably succumb to recurrences (Stupp et al, 2005). Two key features of glioblastoma biology contribute to recurrences: firstly glioblastoma growth is highly infiltrative and sparse at the margins, rendering complete surgical resection impossible and secondly, glioblastoma harbours a population of glioma stem/ initiating cells (GIC), that show elevated therapeutic resistance, are capable of self-renewal, and repopulate the tumour after surgery (Gimple et al, 2019; Lathia et al, 2015).

The clinical benefit of numerous rounds of surgery upon glioblastoma recurrence are unclear, and consequently only up to 25% of patients undergo surgery at recurrence (Weller et al, 2013). However, despite these limitations, several multi-centre initiatives have now brought together large-scale longitudinal cohorts to enable analysis of patient-matched changes in glioblastoma genetics, epigenetics, transcription and cellular composition that occur with recurrence (Barthel et al, 2019; Drexler et al, 2024; Varn et al, 2022; Wang et al, 2022). Bulk transcriptional analysis of glioblastomas identified three subtypes proneural, classical, and mesenchymal (Verhaak et al, 2010; Wang et al, 2017). Each of these are differentially composed of four major cell states that loosely resemble healthy neuroglial cell populations: oligodendrocyte progenitor cell (OPC)-like, neural progenitor cells (NPC)-like, astrocyte (AC)-like and mesenchymal (MES)-like cells (Neftel et al, 2019). Longitudinal analyses indicate that recurrence is often associated with a shift to the poorer prognosis mesenchymal phenotype, a process mediated by AP-1 transcription factors, and driven by paracrine signals from the immune microenvironment (Varn et al, 2022; Wang et al, 2022). Similarly, longitudinal analysis of DNA methylation array-based subtypes, a commonly used tool

[1]Brain Tumour Research Centre, Blizard Institute, Faculty of Medicine and Dentistry, Queen Mary University London, London, UK. [2]Centre for Evolution and Cancer, The Institute of Cancer Research, London, UK. [3]Division of Neuropathology, The National Hospital for Neurology and Neurosurgery, University College London Hospitals NHS Foundation Trust, and Department of Neurodegenerative Disease, Queen Square, Institute of Neurology, University College London, Queen Square, London, UK. [4]Barts Brain Tumour Centre, Faculty of Medicine and Dentistry, Queen Mary University London, London, UK. [5]These authors contributed equally: Sara Lucchini, James G Nicholson. ✉E-mail: s.marino@qmul.ac.uk

for the diagnosis of central nervous system (CNS) tumours (Capper et al, 2018), indicates that subtype-switching occurs at recurrence, most commonly as a mesenchymal shift (Drexler et al, 2024). Results from genomic studies have shown minimal evidence of recurrence-specific gene alterations in these tumours outside of the TMZ-driven hypermutator phenotype (Barthel et al, 2019).

These advances in our understanding of glioblastoma recurrence have yet to be matched by more effective therapeutic approaches. Indeed, numerous therapies with promising pre-clinical data have so far failed to translate to the clinic. This phenomenon is likely due at least in part to unrepresentative disease models (McFaline-Figueroa and Wen, 2023), in particular the widespread use of tumour models of primary (untreated) glioblastoma at pre-clinical level followed by trials targeting recurrent disease. To date, few models of glioblastoma recurrence have been developed and whether they accurately recapitulate clinical progression of glioblastoma at cellular and molecular level after treatment, has not been assessed. Liu et al have developed a recurrence model including surgery and RT/TMZ treatment, but based upon an engineered-murine GIC cell-line G422$^{TN}$-GBM (Liu et al, 2021). Qazi et al used patient-derived human GIC but their treatment course did not include a surgery step, which is the first line of treatment, either for tumour resection or to secure tissue diagnosis if a debulking is not feasible, in glioblastoma patients (Qazi et al, 2022). None of the available models have been validated by comparative analysis with longitudinal samples of glioblastoma recurrence from the same patients.

Here, we present the first glioblastoma recurrence model, in which the standard of care is faithfully replicated and patient-matched longitudinal samples are available to interrogate the patient-specific accuracy with which tumour recurrence is modelled. This allowed us to test the degree to which the true recurrence-associated genomic and phenotypic changes a particular tumour underwent are recapitulated by the model. We show that our induced-recurrence PDX model recapitulates numerous verifiably recurrence-associated phenotypic changes, including the emergence of a population of ciliated neural stem-like cells (cNSC). Furthermore, we demonstrate how scRNAseq or deep exome sequencing of our model can identify druggable, patient-specific therapeutic vulnerabilities of recurrent cells, raising the possibility that it could be used in prospective precision medicine approaches.

## Results

### Establishment of an induced-recurrence xenograft glioblastoma model that recapitulates patient care

Glioblastoma invariably recurs, but few recurrent tumours are operated on, and few (if any) representative models of recurrence exist. Here, we derive GIC from two matched primary-recurrent pairs, GBM39 and GBM67, named GIC (from primary tumour) and GICR (from recurrent tumour), respectively. Both patients received standard-of-care therapy (Radiotherapy (60 Gy/30#) + TMZ followed by adjuvant TMZ), and a summary of their clinical and histo-molecular features is included in Fig. EV1. GIC or GICR were then injected intracranially to generate xenografts that enabled evaluation of how patient tumours changed upon

recurrence, and how well these changes were recapitulated in PDX models (Fig. 1A). In addition, to overcome the lack of glioblastoma tissue at recurrence, we developed a novel induced-recurrence PDX model (IR-PDX) in which PDXs are generated from primary GIC and are then treated with a needle injury—to mimic tissue injury caused by a surgical operation in patients, targeted radiotherapy, and a course of chemotherapy with temozolomide (TMZ), followed by observation until tumours regrow.

Firstly, all GIC lines were injected intracranially into the caudo-putamen of recipient NOD-SCID mice (12-13 mice for each GIC genotype) to generate PDX models of primary and recurrent glioblastoma (Fig. 1A). Prior to intracranial injection early passage GIC and GICR (p2-p7) were stably transduced with a lentivirus carrying the Firefly Luciferase reporter gene to enable continuous in vivo monitoring of tumour growth. We confirmed successful integration in each cell line via IVIS bioluminescence imaging and in vitro luminescence assay (Fig. EV1B). Real-time tumour growth was monitored with in vivo bioluminescence imaging (BLI) (Fig. EV1C), and upon the onset of symptoms, mice were culled and brains harvested for histology or dissociated for the derivation of PDX-derived GIC lines (XGIC). Mice injected with GIC39 showed a median survival of 101 days with tumours displaying two distinct growth patterns: one group with faster disease progression and a second group with slower growth and longer survival (Fig. 1B). By contrast, mice injected with recurrent GIC39R had a shorter median survival of 85 days, with no evidence of a bimodal survival curve (Fig. 1B). For mice injected with either GIC67 or GIC67R the change in median survival was less pronounced, being 114.5 and 120 days, respectively (Fig. 1B). Histological analysis confirmed that PDX tumours were pleomorphic glial neoplasms with high mitotic activity, necrosis and vascular proliferation, and retained expression of human Nestin, in keeping with glioblastoma (Figs. 1C and EV1D,E). Quantification of tumour cells in the brain with a human-specific vimentin (hVim) immunostaing (Fig. 1C) showed non-significant trends towards increased tumour size (percentage of hVim staining area relative to tissue area) in PDXs from GICR for both patients (Fig. 1D) and a reduction in invasiveness (ratio of the gross tumour area to the core area, invasiveness index) (Amodeo et al, 2017; Constantinou et al, 2024) in the GIC39R condition (Fig. 1D), the latter finding in keeping with the histological assessment of compact, well-demarcated tumours seen in these mice (Fig. 1E). Quantification of Ki67 and cleaved caspase 3 immunopositivity showed that proliferation remains constant, and that reduced tumour cell death likely underpins the increased tumour sizes in PDXs from GICR for both patients (Figs. 1D and EV1F).

Next, we set out to optimise the treatment regimen including TMZ and radiotherapy tolerability assays (Fig. EV2A–D) for the IR-PDX model. For these experiments a cell line, GIC80, from a patient with methylated MGMT promoter was chosen (Fig. EV1A), given these tumours are more responsive to TMZ treatment. 18 NOD-SCID mice were orthotopically injected with GIC80 and tumour growth was monitored via IVIS bioluminescence imaging (BLI). Mice were injected intraperitoneally daily with either vehicle alone, 25 mg/kg TMZ, or 50 mg/kg TMZ (with $n = 3$ per group) and brains were collected for analysis at days 5 and 10 (Fig. EV2A). Both doses and schedules of administration were well tolerated by the animals during the time course of the assay. To assess the

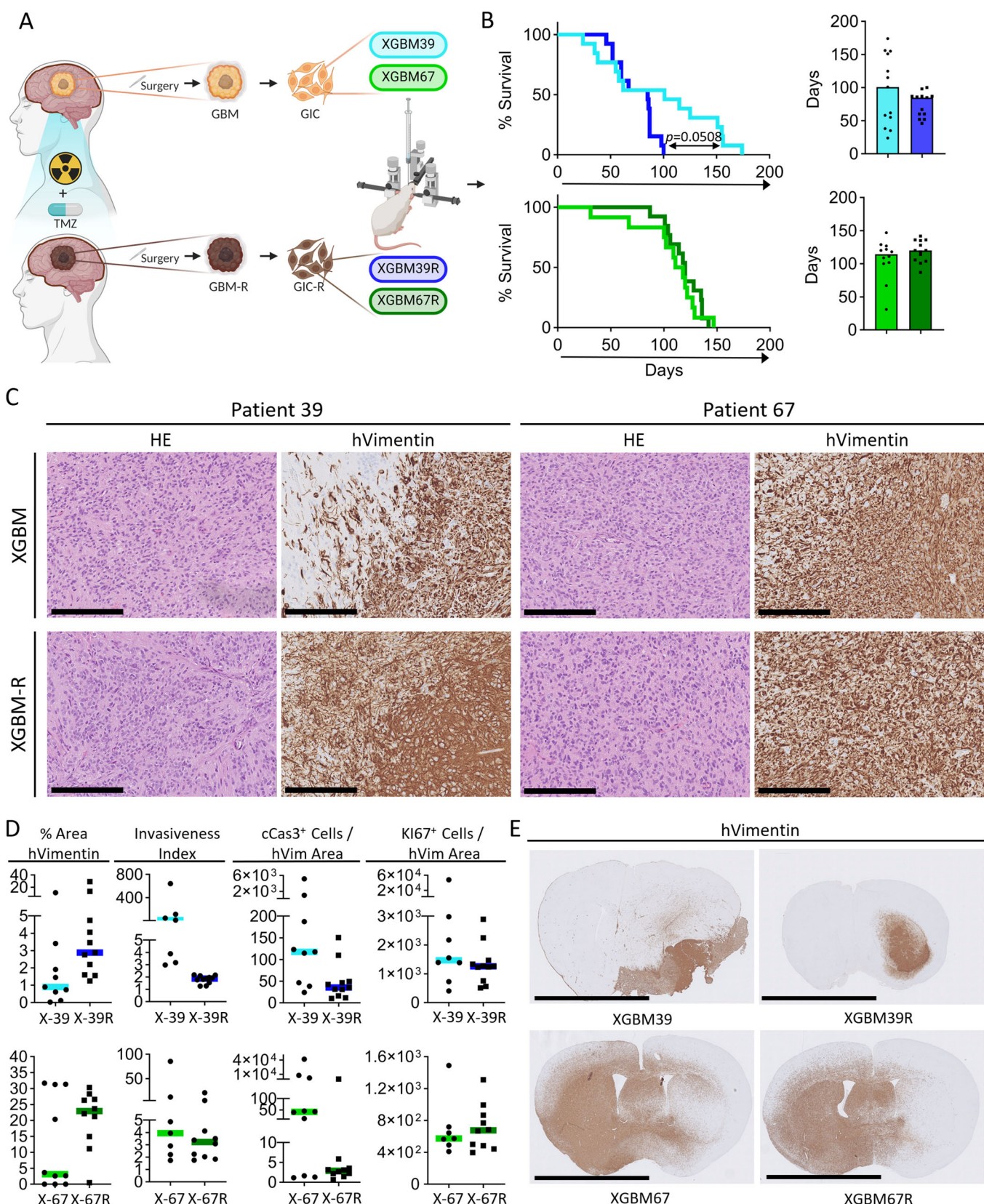

**Figure 1. Characterization of PDXs derived from longitudinal primary and recurrent GICs.**

(A) Schematic of the generation of PDXs models of primary and recurrent glioblastoma derived from patients' longitudinal samples. (B) Survival of PDXs derived from the injection of primary and recurrent GICs; data relative to patient 39 (top), data relative to patient 67 (bottom). Kaplan Meyer graphs, showing the survival of all engrafted mice (left); bar plots showing the median survival (right). $n = 13$ mice for XGBM39, XGBM39R and XGBM67R; $n = 12$ mice for XGBM67. Log-rank (Mantel-Cox) test. Comparison of curves is ns. (C) Representative histology (H&E, left and hVimentin, right) of glioblastoma originating in the PDXs, patient 39 on the left and patient 67 on the right. Top row is showing tumours derived from GICs; Bottom row is showing tumours derived from GICRs. Scale bar corresponds to 250 µm. (D) Column scatter plots showing the quantification of immunohistochemical staining for hVimentin (%area of brain section surface which is positive for hVim and invasiveness index), cCaspase3 and KI67 in XGBM and XGBMR. Top: data relative to patient 39. Bottom: data relative to patient 67. Horizontal line represents the median in all graphs. $n = 9$ mice for XGBM39; $n = 11$ mice for XGBM39R; $n = 10$ mice for both XGBM67 and XGBM67R. All differences are ns. (E) Representative whole brain sections of the PDX models stained for hVimentin. Scale bar corresponds to 5 mm. Source data are available online for this figure.

efficacy of the drug we stained brain sections for $\gamma$-phosphorylated histone 2 AX ($\gamma$H2AX), a marker of DNA double-strand breaks (Collins et al, 2020) induced by TMZ, and for hVim to identify the human tumour cells. Quantification of double positive cells within the tumour and the surrounding tissue established 50 mg/kg for 5 consecutive days as the combination with highest DNA damage in the tumour and lowest toxicity in the surrounding tissue (Fig. EV2A,B). For the radiation tolerability assay, we used a small animal radiotherapy platform (SARRP) to irradiate the forebrain of 5 NOD-SCID mice with a single dose of 10 Gy, whilst avoiding sensitive areas such as brainstem, eyes, oral mucosa, cerebellum and olfactory bulbs (Fig. EV2D). We monitored these mice over 10 weeks and they did not show adverse effects. Next, we generated PDXs for GIC39 and GIC67 lines and monitored in vivo tumour growth via BLI (Fig. EV1B) to identify the timepoint for starting the treatment (Fig. EV2C). To mimic the clinical management of glioblastoma patients, we began our treatment regime at the time when tumours became visible by BLI, at week 1 (patient 39) or week 8 (patient 67). To begin the treatment (Fig. 2A), we induced a needle injury to mimic a biopsy, using the same stereotactic coordinates as the GIC injection. For patient 39, which has a faster time course of tumorigenesis, the injury was induced at week 2, and for patient 67 this was done at week 6. Then, after 7 or 21 days (for patient 39 and 67, respectively) focal radiotherapy (RT) was applied, followed two days later by 5 days of TMZ administration. Tumour size was assessed after the operation and then monitored once a week from the end of the treatment (Fig. 2B,C). Both IR-GBM67 and 39 showed a reduction of signal after treatment, indicating a shrinkage of the tumour mass as compared to untreated XGBM39 and XGBM67, which was followed by tumour recurrence marked by an increased BLI signal (Fig. 2C). Histological assessment of IR-GBM tumours, showed well-demarcated tumours upon hVimentin (Fig. 2D) and hNestin (Fig. EV2G) staining for both patients and H&E confirmed glioblastoma morphology (Fig. 2D).

A clinically useful model of glioblastoma recurrence should recapitulate patient responses to specific therapies. Given the prognostic use of MGMT promoter methylation in glioblastoma treatment, we therefore compared the efficacy of our full treatment regime on PDX implanted with MGMT methylated GIC80 and unmethylated GIC67 or GIC39 cells. For IR-GBM80, where the treatment is expected to be more effective, there was a significant reduction in BLI detected tumour growth when compared to XGBM80 untreated controls (Fig. 2E). Although a similar trend was observed also for both IR-GBM67 and IR-GBM39, the comparison with the untreated XGBM67/XGBM39 did not reach statistical significance (Figs. 2E and EV2E, respectively). Similarly,

quantification of endpoint tumour size (percentage of hVim staining area relative to tissue area), showed a significant reduction in treated PDXs derived from GIC80 but not GIC39 or GIC67 cells (Fig. EV2GF). Quantification of Caspase C3 staining indicates this difference was primarily driven by increased cell death in treated GIC80 PDXs (Fig. EV2F). Together, these results suggest that our approach accurately models the expected differences in patient-specific therapeutic responses based on their MGMT promoter methylation status.

## Genomic characterization of the IR-GBM model uncovers a patient-specific therapeutic vulnerability

To assess the genomic, epigenomic and transcriptional fidelity of our models as compared to their matched primary-recurrent tumours, we performed NGS analyses, on either patient (GBM) or PDX (XGBM) FFPE sections (Whole Exome Sequencing, WES), patient- (GIC) or PDX-derived GIC (XGIC) cultures (RNAseq, DNAme), or freshly dissociated PDX-derived tumour cells (scRNAseq) (Fig. EV3A). For WES data from PDX FFPE samples we first computationally removed contaminating mouse reads, minimizing their contribution to called single nucleotide variations (SNVs) (Fig. EV3B,C). After the removal of mouse reads, all samples retained an exome depth coverage of >80X. Comparison between samples revealed an increase in the total number of detected mutations in the recurrent cells for both patients, and that this was reproduced in the IR-GBM67 sample (Fig. EV3D). SNV and copy number variation (CNV) profiles showed that for all cell lines PDX tumour genomes corresponded well to their parental tumour sample (Fig. 3A–C). In the case of patient 39, recurrence was associated with a loss of NF1(pQ912*) mutation and a gain of a mutation in POLE (p.R762W), losses of chromosomes 4 and 18, and amplifications of chr7. For patient 67, detected CNVs included the common glioblastoma pattern of chr7 gain and chr10 loss and remained stable upon recurrence. CNVs can be inferred using DNA methylation array data, we therefore used the greater number of samples profiled by DNAme to supplement the high confidence WES findings and interrogate inter-sample genomic heterogeneity (Fig. 3C). For patient 39, XGICs retained the characteristic copy number profiles of their parental tumours and patient-derived GIC samples. For patient 67, the major chromosomal aberrations remained constant between samples, however, there was small chromosomal loss in chr3 which may have occurred during cell expansion. Interestingly, there is also a loss of chromosome 6 detected in the GICR, as a well as a proportion of the XGIC67R and IR-GIC67. It is possible this CNA is present in a subclone below the level of detection in the original

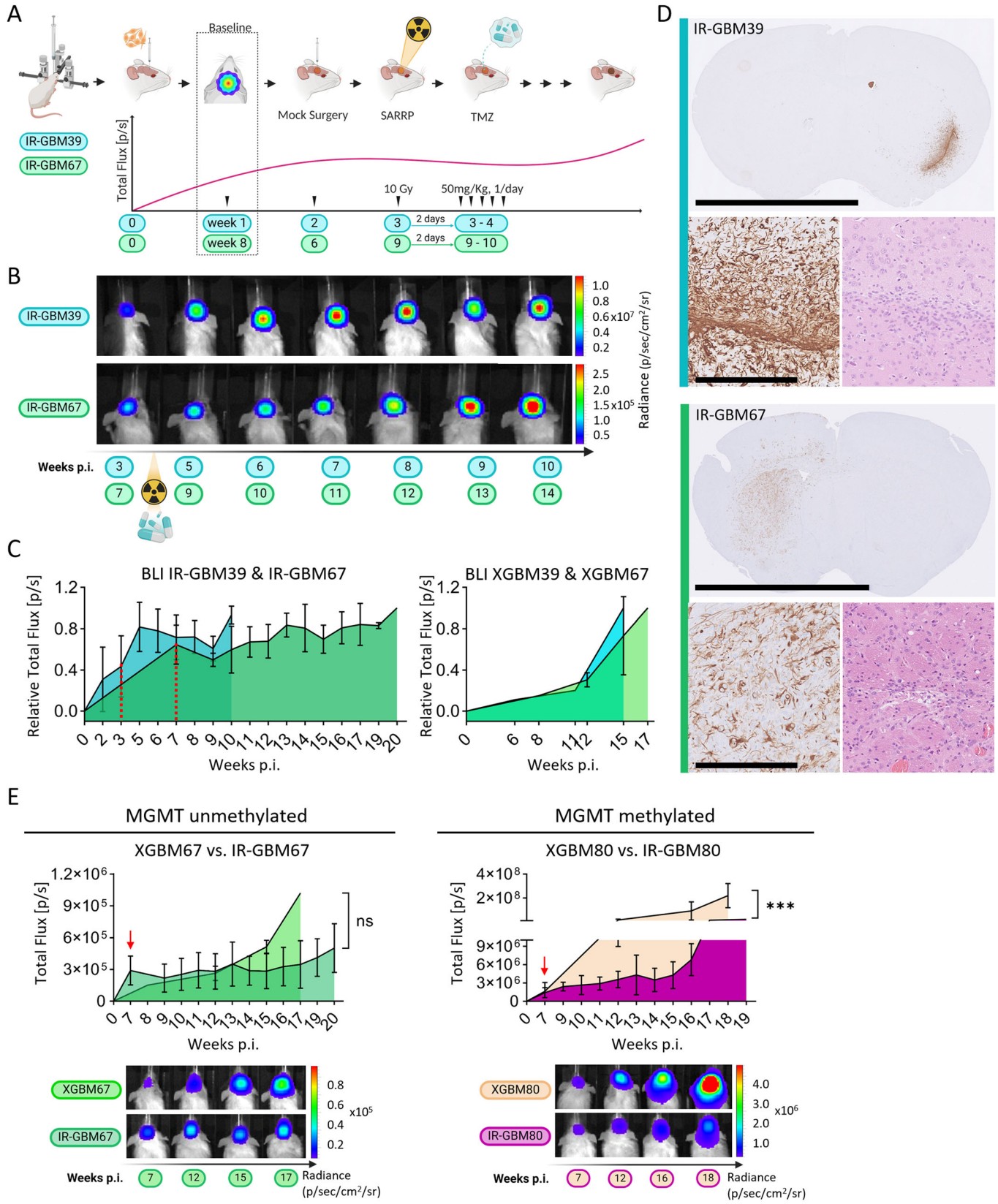

Figure 2. Characterization of a novel induced-recurrence PDX glioblastoma model.

(A) Schematic of experimental design for the patient-specific IR-PDX model. (B) BLI/IVIS monitoring of tumour growth before and after treatment (radiation/TMZ administration). The timeline indicates the week post intracranial injection of primary GICs when the scan was taken. (C) Average tumour growth curves for all mice scanned is shown: IR-GBM39 ($n = 23$) and IR-GBM67 ($n = 4$) (left) and untreated GBM39 ($n = 1$) and GBM67 ($n = 3$) (right). Error bars represent Mean ± SD. Untreated mice were scanned less often. The red lines indicate a timepoint after the sham injection and before radiation and TMZ administration. Each mouse's signal was normalized to the highest signal measured within the same animal over time. (D) Representative histology of the IR-PDX model: IR-GBM39 (top) and IR-GBM67 (bottom). Histology shows hVimentin overview (top) and tumour morphology (hVimentin (bottom left) and H&E (bottom right). Scale bar corresponds to 53 mm for the low magnification and 250 μm for the high magnification. (E) Comparison of tumour growth curves, measured with BLI, between untreated primary PDXs and treated IR-PDXs. Error bars represent Mean ± SD. Top left: growth curves of mice derived by the injection of primary GIC67 ($n = 3$ mice). Top right: growth curves of tumours derived from primary GIC80 ($n = 3$ mice for XGBM80; $n = 4$ for IR-GBM80). The red arrows indicate a first scan performed after the sham injection and before radiation and TMZ administration. Bottom: BLI scans over time of one representative mouse per group of comparison. Unpaired t-test. ***$p = 0.000701$. Source data are available online for this figure.

patient recurrent glioblastoma and variably expanded in different mice.

Notably, analysis of cancer-associated SNVs in the induced-recurrence PDX model detected several low allelic frequency mutations, including the known glioblastoma driver PIK3CA (p.E365K), that were also present in the recurrent GIC67R and XGBM67R but not detected in either of the primary samples for that patient (Fig. 3A). Comparison of all detected SNVs showed the IR-PDX model mimicked a total of 37 mutations detected in the recurrence but not present in the primary tumour (Fig. EV3E). This degree of overlap indicates that the IR-PDX model may be selecting for subclones which would also be selected for during the natural recurrence process in this patient. In accordance with this hypothesis, maximum parsimony phylogenetic analysis of the samples for patient 67 place the induced-recurrence PDX model as an intermediate stage between the primary and recurrence samples (Fig. 3B). Patient-specific genomic insights gleaned from the IR-GBM model may in the future enable prospective precision medicine approaches. To test this possibility, we tested GIC67 and GIC67R for differential sensitivity to Alpelisib, a phosphatidylinositol 3-kinase (PI3K) inhibitor, that selectively acts on mutated PIK3CA (p110α catalytic subunit) (Fritsch et al, 2014) which was found to harbour a gain-of-function mutation in the recurrent and IR model conditions only. Viability of GIC67R was significantly decreased with respect to GIC67 when comparing the AUC of the survival fractions upon treatment for 72 h with increasing concentrations of Alpelisib (Fig. 3D). This result is in keeping with the known effect of PIK3CA (p.E365K) mutation, which is increasing viability and proliferation of the mutated cells. Interestingly, when treating the cells with Idelalisib, a PI3K inhibitor which specifically targets PIK3CD catalytic subunit (p110δ) (Fiorcari et al, 2013), no significant difference in survival between GIC67 and GIC67R was observed (Fig. 3D), confirming the sensitivity of the IR model in identifying and predicting genetic vulnerabilities of recurrence.

## The IR-GBM model recapitulates epigenetic and transcriptional features of recurrence

Glioblastomas are often categorized into one of three transcriptional subtypes (Verhaak et al, 2010; Wang et al, 2017), or four DNA methylation subtypes (Capper et al, 2018), with subtype switching, most commonly towards mesenchymal phenotypes, reported upon recurrence. We therefore checked if any subtype switching occurred in our patients, and the degree to which this was maintained in PDX models (Fig. 4A). Upon recurrence,

GBM39 underwent a mesenchymal transition at both RNA and DNAme levels and, with some inter-sample heterogeneity, this finding was maintained in PDX models. In contrast, GBM67 maintained a GBM IDHwt RTK2 methylation subtype in all samples but had more mixed RNA subtype scores (Fig. 4A). Principal components analyses including all GIC and GBM patient tumour samples separated samples according to their patient genotype and, to a lesser extent, primary or recurrent disease stage, but difference between samples type and preparation (FFPE GBM vs cultured GICs) drove separation over the first component (Fig. EV4A). When principal component analysis of RNA and DNAme data were performed on GIC alone, differences driven by patient genotype and primary or recurrent disease stage were clearer (Fig. 4B). Importantly, for patient 67 from whom we were able to derive IR-GIC, the induced-recurrence cells had acquired both transcriptional and epigenetic features of recurrence and clustered nearer GICR than GIC in PCA space. Similarly, when we performed semi-supervised hierarchical clustering RNA/DNAme data of GIC samples, for both patients and methodologies this separated primary from recurrent GIC (Figs. 4C and EV4A,B). Once again where samples from the IR-GBM model were available, they clustered with PDX generated from recurrent (XGICR) rather than primary GIC (Fig. 4C).

We next queried how well the gene expression and DNA methylation changes that occurred between primary and recurrent GIC (XGIC vs XGICR) were captured by the IR-PDX model (XGIC vs IR-GIC). Cross comparison revealed that the IR-PDX model was able to capture >60% of gene expression changes and >30% of differentially methylated genes between PDX generated from primary (XGIC) or recurrent (XGICR) GIC (Figs. 4D and EV4C). We also considered the overlap between those genes with concordant transcriptional and epigenetic regulation (i.e. hypomethylation and upregulated expression or hypermethylation and downregulated expression) (Figs. 4D and EV4E). This showed that the induced-recurrence model was able to capture aspects of recurrence-associated epigenetic regulation of gene expression. Concordantly regulated genes included ZEB1, a transcription factor and master regulator of epithelial-to-mesenchymal transition, which was hypomethylated and upregulated. Consistent with this change, Gene ontology analysis of the concordantly regulated genes highlighted mesenchyme development as an enriched term (Fig. 4E). Taken together, these analyses show that the IR-GBM model captures transcriptional and epigenetic features of glioblastoma recurrence. Crucially, this suggests that the IR-PDX model can provide prospective information about patient-specific transcriptional and epigenetic changes upon recurrence in a scenario where only primary GIC are available.

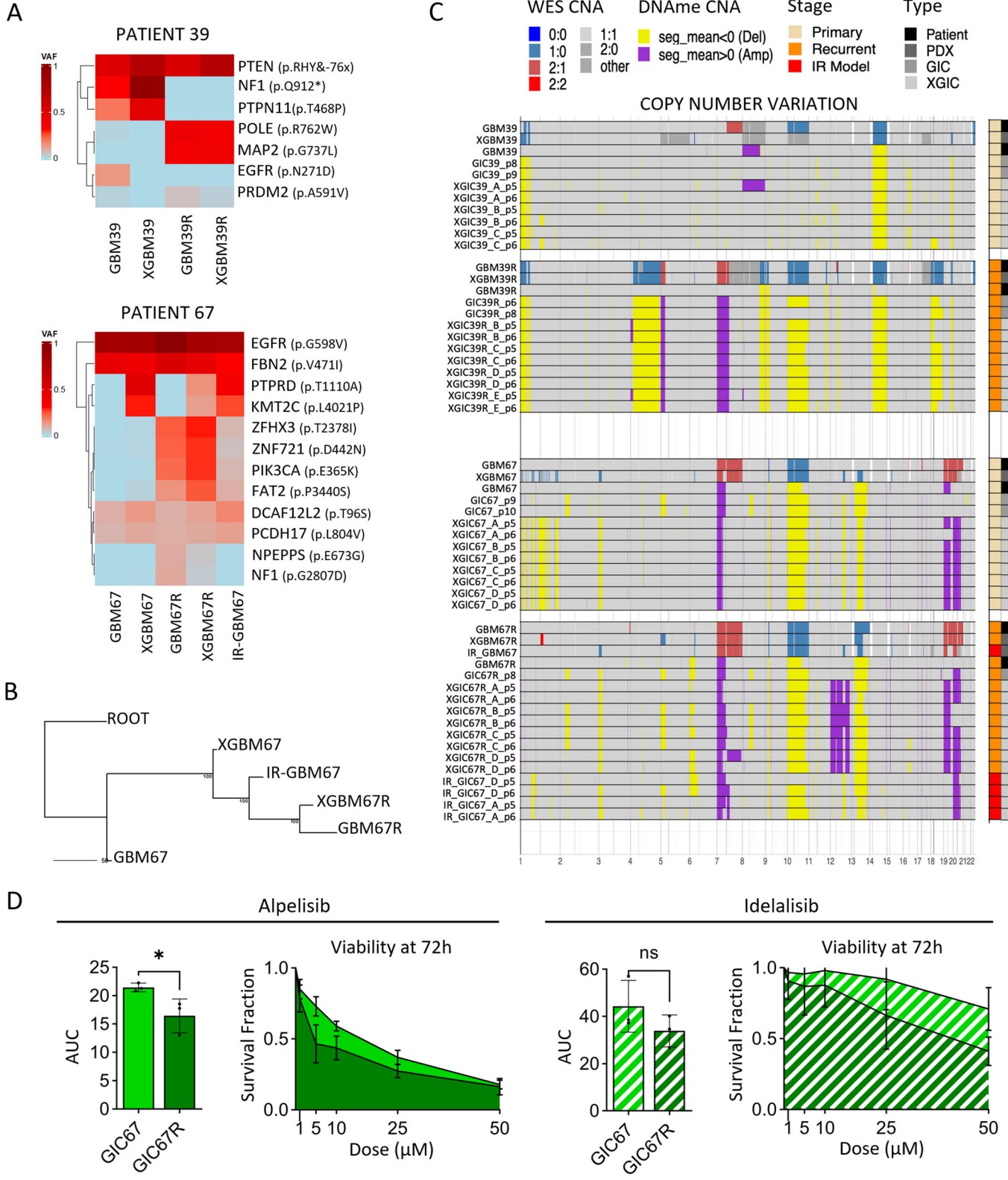

◄ **Figure 3. IR-PDX model accurately recapitulates patient-specific changes in tumour genetics and identifies therapeutic vulnerabilities.**

(A) Heatmaps clustering samples by Variant Allelic Frequency (VAF) at o pan-cancer driver mutations detected in GBM39 (top) and GBM67 (bottom) samples. (B) Maximum-parsimony trees calculating relatedness between GBM67 WES samples. (C) Genomic copy number analysis plots showing genomic fidelity of GIC and GIC PDX models to matched patient samples. Copy number variations calls based on WES data shown in blue (loss) and red (gain) and Copy number variations called based on DNA methylation array data shown in yellow (loss) and purple (gain). (D) Cell viability assay of patient-derived GICs (GIC67) and GICRs (GIC67R), treated with increasing concentrations of Alpelisib (left) and Idelalisib (right) for 72 h. Measurement of Area-under-Curve (AUC) and histograms representing mean AUC ± SD were used to compare the overall response to treatment (bottom), $n = 3$ independent experiments, unpaired t-test. *$p = 0.04$.

## Single cell analysis of the IR-GBM model highlights a recurrence-associated increase in ciliated neural stem-like glioma cells

To evaluate how the treatment course affected recurrence-associated changes in transcriptional state heterogeneity, we performed scRNAseq on primary, recurrence and induced-recurrence PDX samples for both patients, with two replicates per condition. To ensure complete removal of reads originating from contaminating murine cells, we aligned reads to both human (GRChr38) and mouse (Mm10) reference genomes and removed clusters with a mean GRCh38: Mm10 ratio of <3 (Fig. EV5A,B). As expected, gene signature scoring for cell cycle phase indicated that recurrent GIC for both patients were more proliferative than primary GIC, however, the levels of proliferation in the IR-model samples were low, possibly reflecting an earlier stage of tumour recurrence after completion of the treatment regime (Fig. EV5C). Consistent with our bulk and genetic analyses, principal component analysis of the G1-phase cells showed cells from the IR-model adopt an intermediate phenotype between primary and recurrent PDX samples (Fig. EV5D). To compare changes in cell type distributions we performed Louvain clustering and annotated cell type marker genes based on marker gene expression (Fig. 5A–C). For patient 39, PDXs derived from primary cells are predominantly made up of cells expressing glial genes associated with astrocytic and oligodendrocytic identities (APOE, AQP4, BCAN, OLIG1/2). In contrast, PDXs derived from recurrent GIC had undergone a transcriptional shift, with the largest cluster expressing mesenchymal genes (CD44, ANXA2, LGALS3), and a second small mesenchymal cluster expressing high levels of extracellular matrix genes (FN1, BGN, MMP1). In addition to the mesenchymal cells, there were also two clusters that expressed neural stem cell (NSC) genes (NTRK2, HES1, CD24) as well the two major transcription factors associated with primary ciliogenesis (FOXJ1, RFX2), and numerous cilia component genes (CAPS, RSPH1, ARL13B, IFT88). Intriguingly, whilst many cells from the IR-PDX samples retained their AC/OPC-like phenotype, some had transitioned to the recurrence-specific NSC/ciliated states. They also contributed to a cluster of stressed recurrent cells which expressed genes associated with TP53 signalling (CDKNA1, MDM2) and apoptosis (BAX). Gliomas generally, and glioblastoma specifically, are characterized by high interpatient heterogeneity (Nicholson and Fine, 2021), and we did not see extensive overlap between all cluster marker gene set across patients (Fig. 5D), but many of the changes observed in patient 39 were recapitulated in patient 67. Again, primary GIC67 PDX samples were more characterized by glial cells, with two clusters that expressed astrocytic markers (HOPX, APOE, FABP7), and recurrent GIC67 PDX samples contained more mesenchymal cells (FOSB, JUND). However, in terms of cell type distribution the

mesenchymal shift with recurrence was not as large, a finding supported by gene set scoring for the human tumour MES-like signatures (Fig. EV5F). Intriguingly, patient 67 also contained a cluster of cells expressing NSC and cilia marker genes, which was made up of recurrent and induced-recurrent, but not primary GIC. Another recurrence-associated change in patient 67 that was accurately captured by the IR-model was a loss of an NPC-like cluster (SOX2, DDL1/3) that was present in the primary GIC.

Previous studies have shown that transcriptional (Neftel et al, 2019) and epigenetic states (Johnson et al, 2021) in glioblastoma can be influenced, but are not defined by genetics. We therefore performed CNV inference to see whether we could detect CNV changes in the IR PDX model cells that were associated with transcriptional phenotypes (Fig. EV5G). For both patients, primary and recurrent GIC copy number profiles showed relatively little clonal heterogeneity and accurately replicated WES profiles from parental tumours (Fig. 3A). For GBM39, IR-GIC cells clustered alongside primary GIC, whereas for GBM67 the IR-GIC cells were separated between primary and recurrent clusters. Interestingly for GBM39, some of the IR-PDX GIC harboured a regional amplification of chr1 also detected in GIC39R samples, perhaps indicating this population served as a founder clone for the true in vivo recurrence. Taken together though, these data do not support a strong genetic component driving the phenotypic changes in transcriptional state heterogeneity observed in the IR-PDX model.

Excluding proliferative clusters, the greatest similarity between patient clusters was observed in the NSC/ciliated clusters which shared 870 marker genes (Fig. 5D,E). Consistent with our cell type annotations, the top 10 enriched gene ontology terms for the 870 gene intersect were related to cilia assembly and function (Fig. 5F). To determine whether our PDX-derived finding of an association between glioblastoma recurrence and cilia is also found in patient tumours, we scored cells from a reference cohort of 86 primary-recurrent longitudinal samples (GSE174554; (Wang et al, 2022)) for a recurrence-associated cilia (RAC) signature made up of the top 50 most highly expressed genes found within the 870 gene ciliated cluster intersect (Fig. 5G). At the single cell level, we observed many more cells scoring highly (>mean + 2x s.dev) for the RAC-signature in recurrent tumours, and this translated to a significant increase in mean RAC-score in recurrent tumours (Fig. 5F). Lastly, to validate the PDX finding in the patient samples we performed qPCR and IF, showing an increase in cilia-associated gene and protein expression in recurrent tumours (Fig. 5G–I). We validated the expression of four cilia marker genes which were shared between the ciliated cells clusters found in both patients (Fig. 5B,E): C5orf49, ARL13B, IFT88 and RSPH1. These were tested on the RNA extracted from the FFPE bulk tumours and GBM39R showed a fold change of at least 8 times higher than the primary tumour for

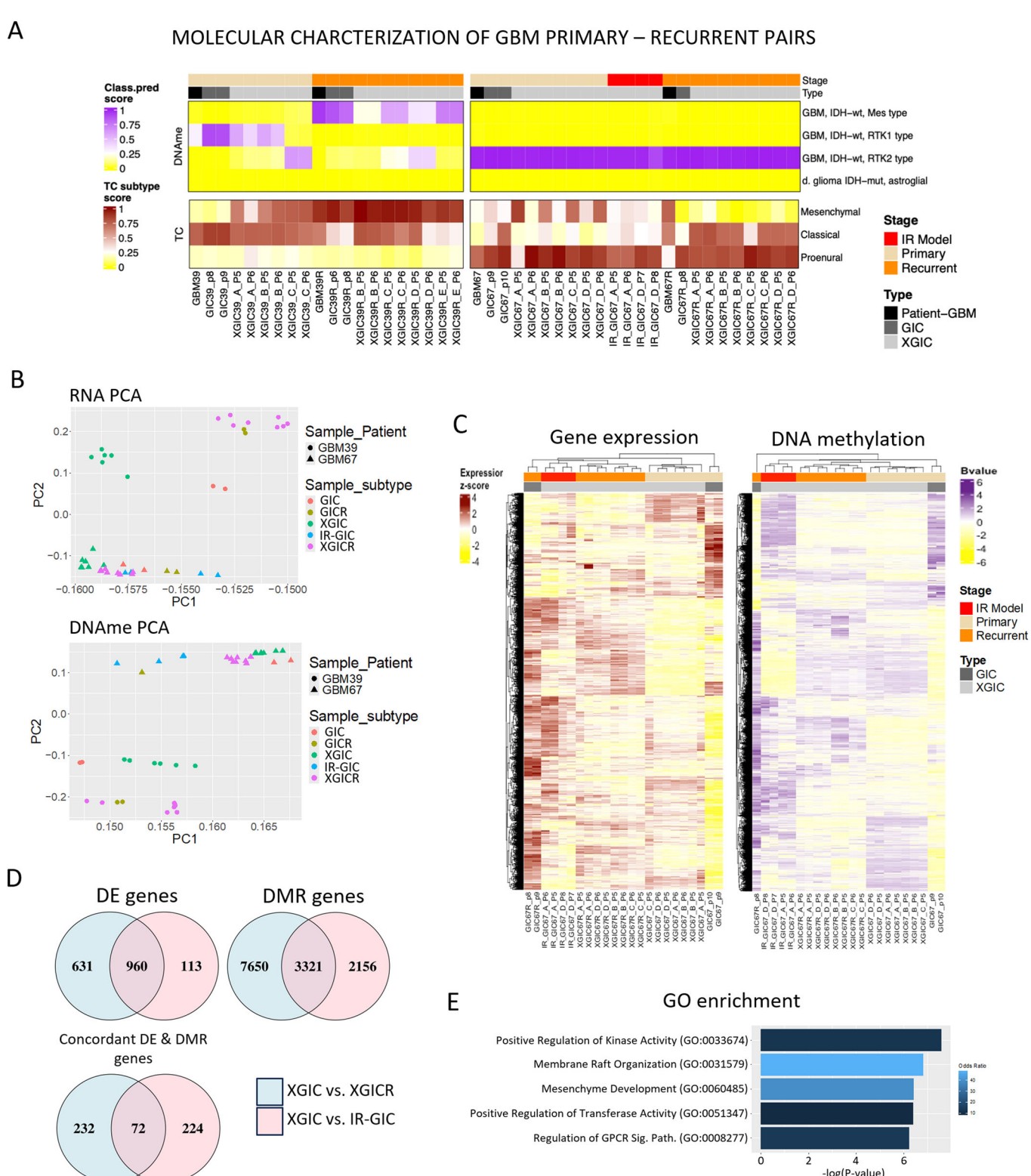

**A** MOLECULAR CHARCTERIZATION OF GBM PRIMARY – RECURRENT PAIRS

**B** RNA PCA

DNAme PCA

**C** Gene expression DNA methylation

**D** DE genes DMR genes

Concordant DE & DMR genes

XGIC vs. XGICR
XGIC vs. IR-GIC

**E** GO enrichment

three out of four genes. Similarly, GBM67R showed increased expression of all four marker genes when compared to the relative primary tumour (Fig. 5G). To confirm this finding at protein-level, we stained primary and recurrent GIC of both patients, for two cilia markers: ARL13B and acetylated α-Tubulin. Both patients showed

a significant increase in cilia expression at recurrence compared to their corresponding primary GICs (Fig. 5H,I).

Previous studies have used genetic reduction (via siRNA against IFT88 (Wei et al, 2022)) or increase (via shRNA against Nek2 (Goranci-Buzhala et al, 2021)) in cilia levels in primary GIC to

◀

**Figure 4. IR-PDX model accurately recapitulates patient-specific epigenetic and transcriptional changes.**

(A) Heatmap of transcriptional and methylation glioblastoma subtype scores. (B) Principal component analysis of RNA samples (top) and DNAme (bottom) patient- and xenograft-derived GIC samples. Samples are coloured according to their subtype (patient-derived primary GICs or recurrent GICRs, xenograft-derived primary XGIC, induced-recurrence IR-GIC or true recurrence XGICR) and shaped according to their patient of origin with patient 39 shown as circles and patient 67 shown as triangles. (C) Semi-supervised hierarchical clustering of RNA (left) and DNAme (right) patient- and xenograft-derived GIC samples. For RNA and DNAme samples, Heatmaps depict the top 5000 variable genes and top 10,000 variable probes (ranked by standard deviation) between GICs, respectively. Semi-supervised hierarchical clustering of RNA samples (left) and DNAme (right) patient- and xenograft-derived GIC samples (D) Venn diagram showing intersections of genes that are differentially expressed, differentially methylated, or concordantly regulated differentially expressed genes (i.e. hypomethylated and overexpressed or hypermethylated and downregulated) (left: blue & pink) genes between XGICs and XGICRs (XGIC vs XGICR; blue) or IR-_XGICs (XGIC vs IR-GIC; pink). (E) Bar plot showing Gene Ontology term enrichment for the intersect of concordantly regulated genes. X axis depicts −log (P-value) calculated in TopGO using a Fisher's exact test.

support their role in mediating TMZ resistance. We therefore queried whether pharmacological inhibition of ciliogenesis with Mebendazole (MBZ (Hong et al, 2024)), could re-sensitize recurrent GIC to TMZ therapy. We first used immunostaining to confirm that 48 h treatment with MBZ caused a dose-responsive reduction in cilia in recurrent GIC (Fig. 6A). We then treated recurrent GIC from both patients with a combination of MBZ and TMZ at varying concentrations (Fig. 6B). For both patients we observed significant synergy between the two drugs.

Finally, we tested whether, in addition to MBZ, we could use scRNAseq data from the IR-GBM model to predict additional recurrence-specific therapies, with a particular focus on patient 39, for which deep exome sequencing did not reveal recurrence-specific targetable mutations. We selected the top 10 most highly upregulated genes shared between primary and induced-recurrence, and primary and true-recurrence cells (Fig. 6C) as candidate target genes, and searched the drug gene interaction database v5 (Cannon et al, 2024) for potential inhibitors. Through this approach we identified Zonisamide a selective SCN4B sodium channel inhibitor for patient 67 as well as Pyrimethamine and TRAM34 for patient 39 which inhibits SLC47A2 (Multidrug And Toxin Extrusion Protein 2) and KCNN4 (Potassium Calcium-Activated Channel Subfamily N Member 4), respectively. In all cases the drug predictions were validated to be more effective in recurrent GIC relative to primary GIC (Fig. 6D).

## Discussion

We developed and characterized a novel induced-recurrence PDX model in which mice xenografted with primary patient-derived GIC undergo treatment analogous to patient standard-of-care before tumours are allowed to recur. Using bioluminescent imaging, we staged our clinical intervention appropriately and confirmed the initial efficacy of treatment prior to recurrence. Crucially, our study incorporated two matched pairs of primary and recurrent GIC, thus enabling for the first time a patient-specific evaluation of a glioblastoma model using the actual recurrent cells as a benchmark. We analysed patient-specific genomic, epigenetic and transcriptional state heterogeneity changes, and showed that at each level the IR-PDX model is capable of recapitulating features of the true recurrence. Patient matched pairs of primary and recurrent GIC are rare, due to very few patients undergoing surgery at recurrence (Weller et al, 2013), hence availability of a predictive model could be transformative in advancing the development of effective second line treatments for glioblastoma.

Longitudinal sequencing studies have shown an absence of common mutations that drive recurrence across a large proportion of patients (Barthel et al, 2019), and the repeated failure of glioblastoma clinical trials are often attributed to its high level of genetic heterogeneity. However, some patients do harbour specific targetable mutations that likely contribute to recurrence at the individual level, and it is likely that insufficient patient stratification has contributed to some of the failures of targeted therapies in glioblastoma. There is therefore increasing interest in applying precision medicine and patient-specific approaches to glioblastoma therapy (Vinel et al, 2021). In this context, it is promising that deep sequencing on the IR-PDX model identified a druggable, patient-specific therapeutic vulnerability of the GIC67R, which would not have been identified in GIC67. In contrast, genetic sequencing of the true recurrent GIC from patient 39 did not reveal newly acquired and targetable mutations. Given the overall paucity of effective targeted therapies in glioblastoma (Fougner et al, 2024), this may often be the case. To overcome this, we instead used available IR-GIC39 scRNAseq data to predict transcriptional based precision medicine targeted therapies. We found both TRAM34 and Pyrimethamine to be more effective against recurrent relative to primary GIC. This suggests that when induced-recurrent modelling does not reveal genetic vulnerabilities, transcription-based drug predictions may be a viable alternative.

Bulk comparisons indicated that the IR-PDX recapitulated ~60% of the transcriptional changes between primary and recurrent GIC, whereas only 30% of DNA methylation changes were detected. This may be due to the relatively slower rate of DNA methylation changes, and the accelerated recurrence in mice (weeks) relative to human patients (months). Nevertheless, the induced-recurrence model still identified more than 3000 genes that were also differentially methylated in the true recurrence, and DNA methylation data can be an important input to glioblastoma precision medicine pipelines (Vinel et al, 2021). Therefore, how well epigenetically or protein-protein-interaction based (Cheng et al, 2019a; Cheng et al, 2019b) predictions compare to those based on genomic and transcriptional data should be further investigated with our model.

Given the propensity of glioblastoma to recur, and the lack of clinical indications to re-operate at recurrence, a major goal for patient-specific therapeutic approaches to recurrence should be to develop predictive models. Here we provide proof of concept that a glioblastoma recurrence model could provide therapeutic insights, which would not be available from the primary tumour cells alone. A strength of our study is the ability to cross-compare induced and true-recurrent predictions, which would be unavailable in a

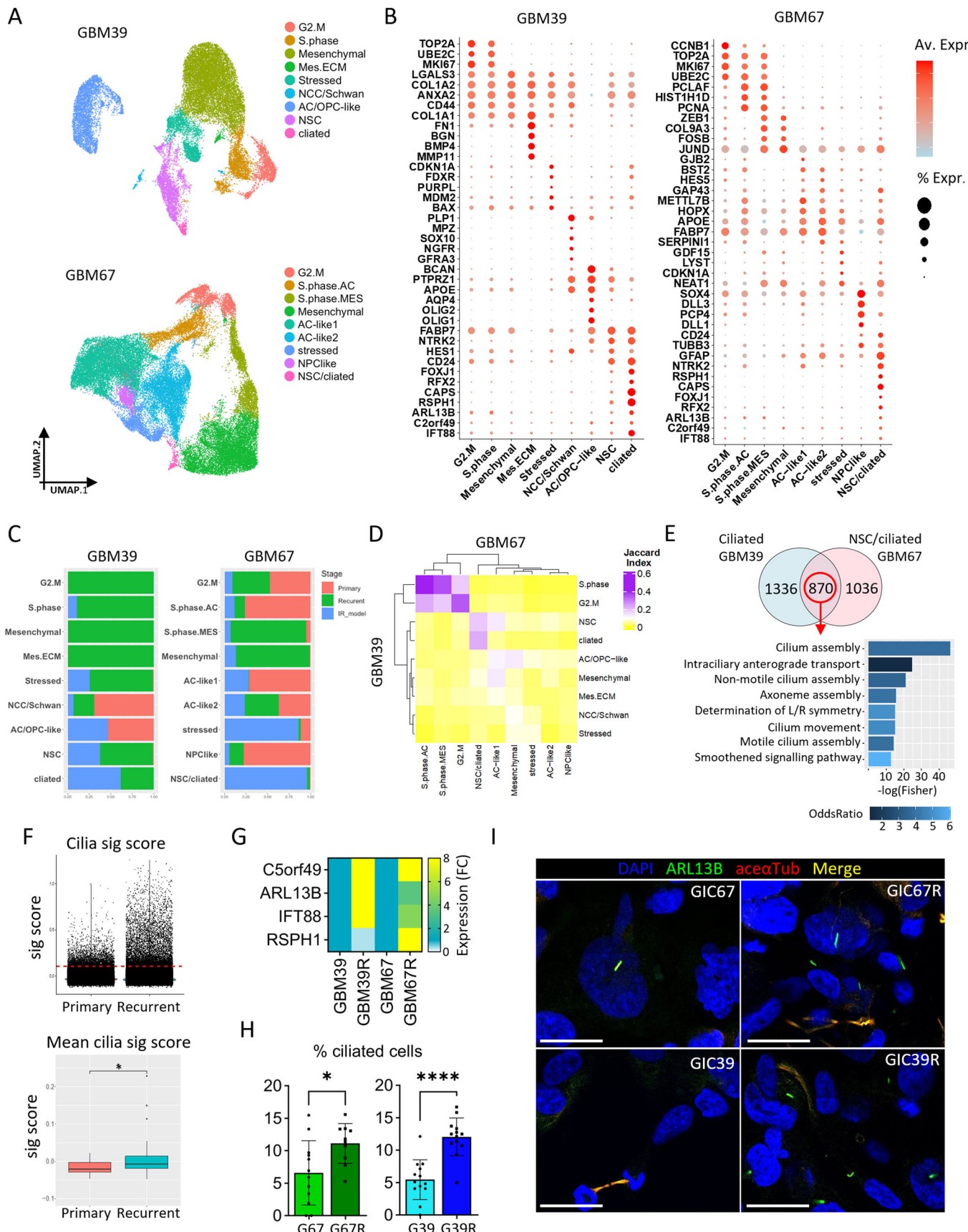

◄ **Figure 5.** **IR-PDX model accurately recapitulates recurrence-associated changes in transcriptional state heterogeneity.**

(A) UMAP of GBM39 (top) and GBM67 (bottom) PDX samples coloured by annotated cell type. (B) Dot plots of cluster marker gene expression used for cell type classification. (C) Bar plots showing the relative proportions of each GIC model in annotated cell types for GBM39 (left) and GBM67(right). (D) Heatmap showing marker gene overlap between GBM39 and GBM67 samples, expressed as a Jaccard index. (E) Venn diagram showing intersection of cluster marker genes between ciliated clusters in GBM39 and GBM67 (top). Bar plot showing the top 8 enriched GO terms for the intersect gene list of the ciliated clusters (bottom). (F) Recurrence-associated cilia signature scoring in a reference cohort of GBM patients, in a violin plot of single cells (top) and a boxplot showing scores averaged by tumour (bottom). The boxplot centre line represents the median (50th percentile), while the lower and upper bounds of the box represent the 25th (Q1) and 75th (Q3) percentiles, respectively. The whiskers extend to the minimum and maximum values within 1.5 × IQR (interquartile range) from Q1 and Q3, and data points beyond this range are considered outliers (depicted as individual points). N = Primary, 30; Recurrent, 41, Wilcoxon test. *p = 0.016. (G) qPCR for cilia marker genes on parental glioblastoma samples (primary and recurrent) from patient 39 and 67. (H) Quantification of fluorescence images with each cilia counted and the total number of cilia within each field normalized to the total number of nuclei within that field. Each field contained 25 tiles, and each tile was captured at 63X magnification. Each cell line was stained at three different passages. n = 12 fields quantified per line. Error bars represent Mean ± SD. Unpaired t-test. *p = 0.021; ****p = <0.0001. (I) Representative fluorescence images of cilia stained for ARL13B and aceαTubulin in the two primary GIC lines (left) and the two recurrent GIC lines (right). Scale bars are 20 μm. Source data are available online for this figure.

scenario in which patient recurrences are treated based on induced-recurrence predictions alone. However, in in-depth understanding of the reproducibility and accuracy of recurrence modelling across patients in a much larger experimental cohorts than presented in our study is needed, to justify taking the approach forward for personalised precision medicine. It can be envisaged, however, that combining therapeutic predictions based on induced-recurrence modelling with testing on GICR in the cases where they are available, could potentially represent a more readily translatable approach.

In addition to the commonly reported mesenchymal shift, our scRNAseq analyses also identified an increase in a small cluster of ciliated cells with NSC-like expression programmes in recurrent cells. Importantly, this recurrence-associated change in cell state heterogeneity was present in both patients, and our RAC-signature was also found to be enriched in recurrent patient samples of a reference cohort (Wang et al, 2022). Intriguingly in sub-ventricular NSC—widely considered as glioblastoma cell of origin (Lee et al, 2018) and GIC's closest healthy homologue—primary cilia distinguish quiescent stem cells from those that are actively dividing (Khatri et al, 2014). This close association between NSC cilia and the cell cycle is at least partially retained in GIC, and loss of primary cilia is an important requirement for the maintenance of a highly proliferative state in GIC (Goranci-Buzhala et al, 2021; Loskutov et al, 2018; Yang et al, 2013). Cancer stem cell quiescence is thought to contribute to drug resistance and recurrence, and quiescent cancer stem cells have been identified in a wide range of tumours, including glioblastoma (Chen et al, 2012). Typical anticancer treatments, including TMZ and radiotherapy, damage DNA, often causing maximal cell death at the point of cell-division. By maintaining $G_0$, quiescent cells can therefore evade chemotherapy-induced cell death, persist under treatment, and re-enter the cell cycle once the course is complete. Consistent with this model, in vitro modulation of cilia levels (via siRNA against IFT88 (Wei et al, 2022), PCM1 (Hoang-Minh et al, 2016) or Nek2 (Goranci-Buzhala et al, 2021)) in primary GICs supports their role in promoting TMZ resistance. We reasoned that the reported links between cilia and TMZ resistance may underpin their association with recurrence, and tested whether pharmacological inhibition of ciliogenesis with Mebendazole (MBZ (Hong et al, 2024)), could re-sensitize recurrent GIC to TMZ therapy. Notably, we found that pharmacological inhibition with MBZ successfully reduced the number of cilia, and synergised with TMZ, increasing recurrent

GIC death in both patients. The potential to increase or prolong the efficacy of TMZ treatment of glioblastoma via pharmacological reduction of cilia is an exciting opportunity for future investigation.

In our bioluminescence tracking of the IR-PDX model we observe an initial reduction in signal and thus tumour burden, followed by tumour regrowth and expansion. It is tempting to speculate that the small population of ciliated NSC-like cells we observe in scRNAseq represent those cells from the original tumour that survived treatment and seeded the recurrence. Since we did not treat mice xenografted with GICR, their increase in cilia was presumably originally induced by the treatment of the patient. Given the known plasticity of glioblastoma transcriptional states (Neftel et al, 2019), the preservation and persistence of this phenotype long after the culmination of patient therapy is noteworthy. The epigenetic mechanisms that maintain the ciliated NSC state through cell line derivation, in vitro expansion and xenografting warrant further investigation. This is particularly true if one considers that in addition to their role in cell cycle regulation, primary cilia also serve as signalling hubs and harbour numerous receptor tyrosine kinases. The role of the smoothened signalling pathway, and its dysfunction in medulloblastoma is the best understood, but cilia also present on their membrane several signalling proteins coded by key oncogenes which are frequently amplified and dysregulated in glioblastoma including KRAS, EGFR and PDGFR (Christensen et al, 2012; Christensen et al, 2017). Intriguingly, cilia have been shown to play a prominent role in acquired and de novo resistance to a diverse range of kinase inhibitors and cancer cell lines, and drug sensitivity could be rescued by genetic or pharmacologic ablation of cilia (Jenks et al, 2018). Despite much enthusiasm, as a class receptor-tyrosine kinase inhibitors have performed poorly in glioblastoma clinical trials, a putative role for cilia in mediating resistance to these drugs should be further explored.

The lack of accurate pre-clinical models is an often-cited challenge contributing to the lack of advancement in glioblastoma therapies (Fine, 2024; McFaline-Figueroa and Wen, 2023). Using multi-modal sequencing and longitudinally matched GIC pairs, we demonstrate that our IR-PDX model can faithfully replicate some patient-specific phenotypes associated with recurrence, yielding biological insight and identifying therapeutic vulnerabilities and could therefore represent a useful tool for the glioblastoma research community.

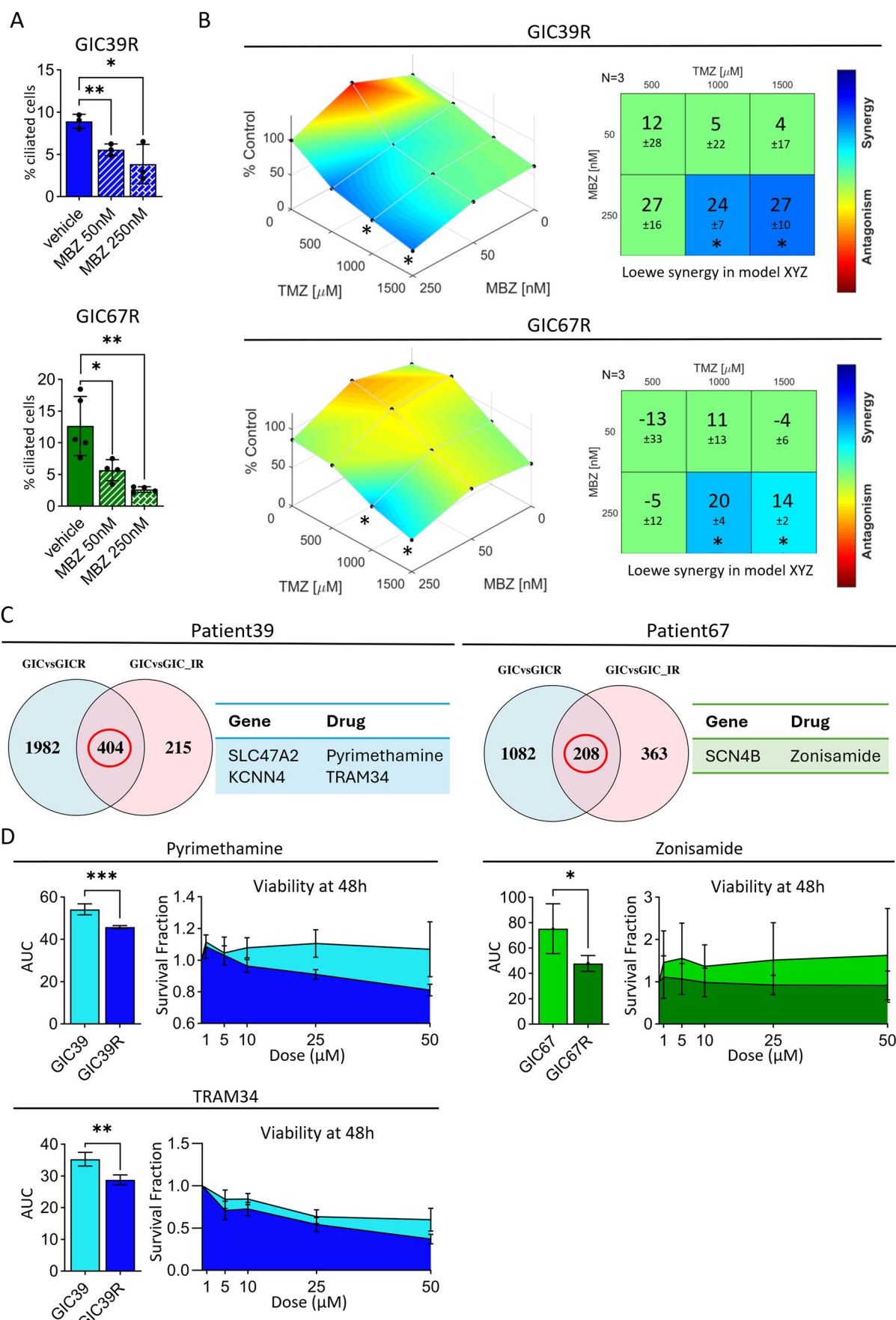

**Figure 6. Transcriptional analysis of the IR models highlights candidate genes for targeted therapeutics.**

(A) Quantification of fluorescence images after cilia inhibition. Both cell lines were either treated with DMSO alone, 50 nM and 250 nM of MBZ for 48 h before fixation. Each cilia was counted and the total number of cilia within each field normalized to the total number of nuclei within that field. Each captured field contained 25 tiles, and each tile was captured at a 63X magnification. $n = 3$ fields quantified per condition, data shown as mean ± SD. Unpaired t-test. GIC39R: *$p = 0.0242$ and **$p = 0.0057$; GIC67R: *$p = 0.0255$ and **$p = 0.0039$. (B) Left: surface plots of synergy scores obtained after combination treatment of MBZ and TMZ on recurrent GIC (top: Patient 39; bottom: Patient 67). $n = 3$ biologically independent experiments. Right: matrices reporting Loewe Synergy Scores obtained after combination treatment of MBZ and TMZ in GIC39R (top) or GIC67R (bottom). $n = 3$ independent biological experiments. (C) Venn diagram showing intersections of differentially expressed genes between XGIC and XGICR or IRGIC—samples processed for scRNAseq. Patient 39 on the left and patient 67 on the right. Both intersect presented a list of potential targets. Genes selected as potential recurrence patient-specific targets from the top 10 DE genes and the corresponding drugs which were selected are shown next to the relative Venn diagram. (D) Cell viability assay of patient-derived GICs and GICRs, treated with increasing concentrations of Zonisamide (right, patient 67), Pyrimethamine (top left, patient 39) and TRAM34 (bottom left, Patient 39). Measurement of Area-under-Curve (AUC) and histograms representing mean AUC ± SD were used to compare the overall response to treatment. $n = 4$ independent biological experiments, unpaired t-test. *$p = 0.037$ (Zonisamide); **$p = 0.0028$ (TRAM34); ***$p = 0.0008$ (Pyrimethamine).

# Methods

### Reagents and tools table

| Reagent/ Resource | Reference or Source | Identifier or Catalog Number |
|---|---|---|
| **Experimental models** | | |
| GIC39 | NRES 08/H0716/16 Amend.1 17/10/14 (this study) | N/A |
| GIC39R | NRES 08/H0716/16 Amend.1 17/10/14 (this study) | N/A |
| GIC67 | NRES 08/H0716/16 Amend.1 17/10/14 (this study) | N/A |
| GIC67R | NRES 08/H0716/16 Amend.1 17/10/14 (this study) | N/A |
| GIC80 | NRES 08/H0716/16 Amend.1 17/10/14 (this study) | N/A |
| NOD SCID CB17-Prkdcscid/J | Charles River UK Limited | N/A |
| **Recombinant DNA** | | |
| Firefly-Luciferase fluorescence reporter | Gift of Prof. Chesler, Institute of Cancer Research | N/A |
| **Antibodies** | | |
| Rabbit anti-hVimentin (IF) | Invitrogen | MA5-16409 |
| Rabbit anti-γ-H2AX | Cell Signaling | #9718 |
| Rabbit anti-ace-α-Tubulin | Cell Signaling | #5335 |
| Mouse anti-ARL13B | NeuroMab | 75-287 |
| **Oligonucleotides and other sequence-based reagents** | | |
| GAPDH-Fwd | Sigma | CTGAGGCTCCCACCTTTCTC |
| GAPDH-Rev | Sigma | TTATGGGAAAGCCAGTCCCC |
| C5orf49-Fwd | Sigma | AATTTCCCACAGTTTAGCAG |
| C5orf49-Rev | Sigma | ATACTTAGGAGTCACTCAGC |
| ARL13B-Fwd | Sigma | GAGCATGAGCAAATAGAGAC |

| Reagent/ Resource | Reference or Source | Identifier or Catalog Number |
|---|---|---|
| ARL13B-Rev | Sigma | TTAGCTGATTCCAATGATGG |
| IFT88-Fwd | Sigma | ATGTCTGCGTTTCTTAGTTC |
| IFT88-Rev | Sigma | CAACCTCTTCAGTTTTCTGG |
| RSPH11-Fwd | Sigma | GGAAAGAGGAGAAGAGGAAG |
| RSPH11-Rev | Sigma | AATTCAGTGATTTGGGTAGC |
| **Chemicals, Enzymes and other reagents** | | |
| DPBS | Corning | 21-030-CV |
| OCT | Fisher Scientific UK Ltd | 12678646 |
| Accumax | Sigma | A7089 |
| Accutase | Sigma | SCR005 |
| murine EGF (20ng/ml) | Prepro Tech | 315-09 |
| human FGF (10ng/ml,) | Prepro tech | AF-100-18B |
| Stem Cell Banker | Ambsio | 11890 |
| NeuroCult™ NS-A Basal Medium (Human) | StemCell Technologies | #05750 |
| NeuroCult™ SM1 Neuronal Supplement | StemCell Technologies | # 05711 |
| Laminin | Sigma | L2020-1MG |
| Laminin | Bio-Techne | 3446-005-01 |
| Hygromycin | Merck | H7772-250mg |
| D-luciferin potassium salt | Promega | E16030 |
| Alpelisib | MedChemExpress | BYL-719 |
| Idelalisib | MedChemExpress | CAL-101 |
| Zonisamide | MedChemExpress | HY-B0124 |
| Pyrimethamine | MedChemExpress | HY-18062 |
| Mebendazole | MedChemExpress | HY-17595 |
| TRAM34 | Tocris | #2946 |
| Temozolomide | Sigma-Aldrich | T2577 |
| DMSO | Santa Cruz Biotechnology | sc-358801 |
| Propofol | Virbac | Propofol®-Lipuro Vet |

| Reagent/ Resource | Reference or Source | Identifier or Catalog Number |
|---|---|---|
| Medetomidine | Orion Pharma | Dormitor |
| Fentanyl | Martindale Pharmaceuticals | PL 00156/0038 |
| Atipamezol | Dechra | Antipam® |
| Butorphanol | Zoetis | Torbugesic® |
| isoflurane | Abbott Laboratories | IsoFlo® |
| Dako Target Retrieval Solution | Agilent Technologies | S236984-2 |
| SuperScriptIII | Invitrogen | 18080093 |
| Sybr Green PowerUp Master Mix | Applied Biosystems | A25742 |
| **Software** | | |
| ImageJ | https://imagej.nih.gov/ij/index.html | N/A |
| GraphPad Prism 9.0c | https://www.graphpad.com | N/A |
| QuPath | https://qupath.github.io/ | N/A |
| BioRender | https://biorender.com | N/A |
| Combenefit software | http://sourceforge.net/projects/combenefit | N/A |
| Living Image 4.3.1 software | PerkinElmer | N/A |
| Cell Ranger 2.0.1 | https://www.10xgenomics.com | N/A |
| R version 4.3.0 (2023-04-21) | https://www.r-project.org | N/A |
| **Other** | | |
| RealTime-Glo™ MT Cell Viability Assay | Promega | G9713 |
| Brain Tumour Dissociation Kit (P) | Miltenyi Biotec | 130-095-942 |
| Mouse Cell Depletion Kit | Miltenyi Biotec | 130-104-694 |
| Debris Removal Solution | Miltenyi Biotec | 130-109-398 |
| FFPE RNA/DNA Purification Plus Kit | Norgen | 54300 |
| RNA/DNA/ Protein Purification Plus Kit | Norgen | 47700 |
| Genomic DNA Clean & Concentrator-25 kit | Zymo research | D4011 |
| Chromium Next GEM Single Cell 3' Kit v3.1 | 10X Genomics | 1000268 |

| Reagent/ Resource | Reference or Source | Identifier or Catalog Number |
|---|---|---|
| Chromium Next GEM Chip G Single Cell Kit | 10X Genomics | 1000120 |
| Human Comprehensive Exome kit | Twist Biosciences | 102032 |
| Plate reader | Synergy HT | 198860 |
| IVIS III imaging system | PerkinElmer | N/A |
| NanoZoomer S360 slide scanner | Hamamatsu Photonics | N/A |
| Cryostat | Leica | N/A |
| SUPERFROST slides | Thermoscientific | J1800AMNZ |
| Zeiss 880 Laser Scanning Confocal microscope | Zeiss | N/A |
| StepOne Real-Time PCR System | ThermoFisher | N/A |

## Human GIC culture

After obtaining informed consent and ethical approval (National Health Service, Health Research Authority, National Research Ethics Service 08/H0716/16 Amendment 1 17/10/2014. All experiments conformed to the principles set out in the WMA Declaration of Helsinki and the Department of Health and Human Services Belmont Report), GICs were isolated from bulk tumour surgical specimens as previously described (Vinel et al, 2021) and according to a published protocol (Pollard et al, 2009). Briefly, GBM tissue was sliced and triturated with a razor blade, dissociated with Accumax (Sigma, A7089) at 37 °C for 10 min then filtered through a 70 μm cell strainer. GICs were maintained on laminin-coated 6-well plates in Neurocult medium plus neurocult supplement (StemCell Technologies, 05750), Pen/Strep, Heparin, murine EGF (20 ng/ml, Prepro Tech, 315-09) and human FGF (10 ng/ml; Prepro tech, AF-100-18B). When 70–80% confluent, cells were detached using Accutase (Sigma, SCR005) and either passaged with a ratio of 1:2 to 1:3 or frozen using Stem Cell Banker (Ambsio, 11890) and stored in liquid nitrogen. All cell lines were regularly tested for Mycoplasma infection and for STR cell authentication profiling. Additionally, strips of dura mater were collected during surgery and used to establish cultures of fibroblasts, both for use as germline controls for genomic analyses in this study, and for future derivation of iPSCs (Vinel et al, 2021).

For introduction of the Firefly-Luciferase fluorescence reporter $0.5 \times 10^5$ GICs were plated in a 24-well plate and then infected using a lentiviral vector inducing the expression of Fluc gene (a gift of Prof. Chesler, Institute of Cancer Research). 250 μl of the supernatant from the packaging cells was both centrifuged at $800 \times g$ for 30 min and filtered with 45 μm pore-sized filters and then added in each well for an overnight incubation. The next day

medium was replaced with fresh culture medium. GICs were subsequently selected with media containing 125 µg/mL hygromycin for one week before further culture or transplantation. Labelled cells were tested for the FLuc expression by adding 150 µg/ml of D-luciferin potassium salt (Promega, E16030) in the culture media and they were immediately analysed using a plate reader (CLARIOstar, BGM Labtech) able to detect Luminescence with a kinetic mode for 60 intervals of 1 s each.

GIC Viability Assay: Cells were treated with Alpelisib (MedChemExpress, BYL-719), Idelalisib (MedChemExpress, CAL-101), Zonisamide (MedChemExpress, HY-B0124), Pyrimethamine (MedChemExpress, HY-18062) and TRAM34 (Tocris, #2946) previously dissolved in DMSO (Santa Cruz Biotechnology, sc-358801) to five concentrations: 1 µM, 5 µM, 10 µM, 25 µM, 50 µM. DMSO has been used alone as vehicle control. Cells were treated with increasing concentrations of drug a day after seeding. Cells viability upon treatment of all drugs apart from TRAM34 was measured real time 1 h after drug exposure for a baseline measurement and then every 24 h for three consecutive days (72 h assay) using RealTime-Glo™ MT Cell Viability Assay (Promega, G9713) as per manufacturer's instructions. Luminescence was measured using a plate reader (Synergy HT, 198860) with integration time 0.5 s. Due to chemical cross-reaction between TRAM-34 and the RealTime-Glo™ MT reagents, cells viability upon TRAM-34 treatment was conducted by manual counting of two technical replicate wells before adding the drugs, and every 24 h for three consecutive days. Viability was calculated normalising the signal of treated cells at 72 h with the one coming from the corresponding vehicle control.

GIC synergy assay: cells were treated with Mebendazole (MedChemExpress, HY-17595) and Temozolomide (Sigma-Aldrich, T2577) either alone or in combination (MBZ doses: 50 nM and 250 nM; TMZ doses: 500 µM, 1000 µM and 1500 µM) and the treatment efficacy was measured using RealTime-Glo™ MT Cell Viability Assay (Promega, G9713) as described above. Synergistic effect of the combined treatment was identified by Loewe Model with Combenefit software (Di Veroli et al, 2016).

## Animal procedures

All procedures were performed in accordance with licenses held under the UK Animals (Home Office Guidelines: animals Scientific Procedures Act 1986, PPL 70/6452 and P78B6C064 Scientific Procedures) Act 1986. The NOD SCID CB17-Prkdcscid/J mice were maintained in individually ventilated cages (IVC), with cage enrichment such as Aspen wood chips, shredded tissue and "fun/handling tunnels". All mice were fed 5R58 diet (PicoLab®) Irradiated during the production process, which was available ad libitum as well as drinking water under a 12-h light/dark cycle at 21.5 °C (±5 °C) and 55% (±10%) humidity. Each experimental group was represented by comparable numbers of male and female animals.

Intracranial injections of GICs: NOD SCID CB17-Prkdcscid/J mice between 10 to 15 weeks-old were anesthetised through intraperitoneal (IP) injection with a cocktail of anaesthetic/analgesic drugs (Propofol: 60 mg/kg, Medetomidine: 1 mg/kg, Fentanyl: 200 µg/kg). Once mice reached a deep sleep state we proceeded with the intracranial (IC) injection of GICs. $5 \times 10^5$ cells were resuspended in 10 µl of PBS and slowly injected with a 26-

gauge Hamilton syringe needle into the right cerebral hemisphere (stereotaxic coordinates from the bregma suture: 2 mm posterior, 2 mm lateral, 4 mm deep, 10° angle). Each mouse was then sutured using 4-0 Coated Vicryl Suture (Ethicon) and then subcutaneously injected with an antidote solution (Atipamezol: 1 mg/kg, Butorphanol: 2 mg/kg; diluted in PBS) able to reverse the anaesthetic effect. Mice were allowed to recover from the surgery on a heatmap until fully awake. Post-operative checks were performed twice daily for seven days, and body weight was monitored once a week thereafter until appearance of first symptoms. For survival experiments, mice were kept on tumour watch until symptoms developed and they were euthanized by neck dislocation. Neurological symptoms due to pathology of the CNS in mice can be represented by: circling; holding the head on one side; disturbance of equilibrium; hunchback position; hypersensitivity upon touching and handling; decrease in food intake immediately followed by weight loss; separation from cage mates; disturbance of eye movements; paresis of a single limb. Bioluminescence analysis: For a proportion of mice, tumour formation and growth was monitored by bioluminescence imaging (BLI) using an IVIS III imaging system (PerkinElmer). A first baseline was performed 6/7 weeks after transplantation and a second check after another 4 weeks. Mice were IP injected with 10 µl per gram of body weight of 15 mg/mL D-Luciferin (Promega, E16030) in DPBS (Gibco) and anaesthetised with isoflurane (Abbott Laboratories) after 5 min from the injection. Images were acquired between 5 and 10 min after D-Luciferin injection, luminescence was recorded with a series of 5 scans of three-minute each using large binning. Total flux (p/s) was measured with Living Image 4.3.1 software using the ROI tool.

Induced-recurrence PDX model:

1. Assessment of the patient-specific treatment time course. The time course of the whole treatment was based on when the specific cell-line showed a first Luciferase signal upon in vivo engraftment.

2. Mock surgery step: at the appropriate time point, mice were anaesthetised as above described, and once they reached a deep sleep state, they were positioned into a stereotaxic frame, and a needle injury (in-and-out without ejection) was performed in the same site as the previous GIC intracranial injection (stereotaxic coordinates from the bregma suture: 2 mm posterior, 2 mm lateral, 4 mm deep, 10° angle).

3. SARRP radiotherapy: Firstly, we performed a pilot study, where 5 NOD SCID CB17-Prkdcscid/J mice were treated using SARRP with a single dose of 10 Gy by irradiating the forebrain with an arc beam. All sensitive areas were avoided (cerebellum, olfactory bulbs, oral mucosa, brainstem, eyes) and mice were closely monitored for adverse effects for 10 weeks. The only observed side effect was a transient fur loss on top of the head. Based on pilot results, the full treatment schedule used for the IR-PDX model was a focal irradiation of the forebrain using SARRP at a 10 Gy dose. During the procedure, mice were blocked in a safe position inside the rotating bed, with the head position being maintained as flat as possible with tape. Mice were anaesthetised with gas anaesthesia (isoflurane) and allowed to recover in a heating chamber.

4. TMZ treatment: First we performed a pilot study, where 18 NOD SCID CB17-Prkdcscid/J mice were IP injected with a solution containing either 25 mg/kg or 50 mg/kg of TMZ (Sigma, T2577)

or the vehicle alone. TMZ was dissolved in DMSO (Santa Cruz Biotechnology, sc-358801) and then diluted in a final vehicle of 5%DMSO, 30% PEG300 and 65% ddH$_2$O. Based on pilot experiments results, the full treatment schedule used for the induced-recurrence PDX model allowed two days of recovery after SARRP radiation and followed by 5 consecutive days of drug treatment with 50 mg/kg of TMZ.

Brain Dissociation and Mouse Cells Depletion: for the derivation of post-PDX GIC lines, mice were culled by neck dislocation and dissected brains and were weighed on a precision scale before dissociation using the Brain Tumour Dissociation Kit (P) (Miltenyi Biotec, 130-095-942) according to the manufacturer's instructions. After complete tissue dissociation, we used the Mouse Cell Depletion Kit (Miltenyi Biotec, 130-104-694) as per manufacturer instructions with minor adaptations, using 100 µL of Mouse Cell Depletion Cocktail and two LS columns (up to $2 \times 10^9$ total cells) per brain, before re-suspension of XGICs in Neurocult medium plus Neurocult supplement (StemCell Technologies, 05750). The day after seeding, the medium was removed and cells were washed twice with PBS (Gibco) in order to remove all debris residues. For the preparation of scRNAseq samples, mouse brains were processed as above, with the additional usage of Debris Removal Solution (Miltenyi Biotec, 130-109-398) as per protocol instructions, between the dissociation and mouse cell depletion steps.

## Image analysis

### Immunohistochemstry

Mouse brains for the pathology analysis were dissected and immediately fixed with 4%PFA at 4 °C and then paraffin embedded. Slides stained with H&E and for hVIM, Ki67, and cleaved Caspase 3 were scanned at 40× objective magnification using a Hamamatsu NanoZoomer S360 slide scanner. QuPath (Bankhead et al, 2017) was used to analyse these whole-slide images (WSI). Colour deconvolution parameters were established based on the highest and lowest intensity of haematoxylin and DAB staining for each stain. Tissue sections were then segmented and annotated using a pixel classifier to distinguish the sections from glass. Subsequently, "Tissue" annotations were eroded by 35 µm to exclude most of the extracerebral tumours. For hVIM images, hVIM detection objects were created using a pixel classifier on the DAB deconvolved channel. "Tumour core" annotations were created by eroding hVIM detection objects by 10 µm to smooth out tumour processes, followed by a 10 µm dilation to restore the original border size. "Tumour core" annotations with an area <10,000 µm$^2$ were removed to exclude small islands. "Gross tumour" annotations were created by dilating hVIM detection objects by 75 µm to fill gaps between close objects, and then eroded by 75 µm to return to the original border distance. "Gross tumour" annotations without a "Tumour core" annotation were excluded. The resulting areas for "Tumour core" and "Gross tumour" annotations were measured and the following calculated: (a) % hVIM area = total area of hVIM/total area of Tissue annotations and (b) invasiveness index = gross tumour area/tumour core area. For Ki67 and cleaved Caspase 3 images, DAB-positive cells were detected using StarDist (Schmidt et al, 2018) from the DAB deconvolved channel. An object classifier was used to identify Ki67- or cleaved Caspase 3-positive cells for

their respective images. The positive cell density within tumour regions was calculated by: Cells per mm$^2$ vimentin = number of positive cells/total hVIM area.

### Immunofluorescence

Mouse brains for the pilot study were dissected and immediately fixed with 4%PFA at 4 °C for 24 h. Subsequently, a 30% sucrose solution was used for dehydrating the tissue, which were left at 4 °C for other 24 to 48 h. Each brain was coronally cut in two parts in the proximity of the injection site and OCT embedded. Each brain was cryosectioned at 10 µm with a Leica cryostat. Slides were left to air dry and then stored at −80 °C. Brain sections were permeabilized and blocked for 1 h at room temperature, with 2.5% BSA, 5% Normal Donkey Serum and 0.1% Triton-X100. Adjacent sections were then stained for hVimentin (Invitrogen, MA5-16409, used 1:200) or for γ-H2AX (Cell Signaling, #9718, used 1:250). For γ-H2AX staining, an antigen retrieval step was performed prior the blocking/permeabilization step. Here, slides were submerged in already boiling antigen retrieval buffer (Dako Target Retrieval Solution, Agilent Technologies, S236984-2) and they were let to gently boil inside it for 5 min. Slides were then let to cool down to room temperature within the buffer. Once at RT, they were washed once with PBS prior proceeding with standard staining protocol (see above). Stained slides were scanned with 40× objective magnification using a Hamamatsu NanoZoomer S60 Digital Fluorescent Slide Scanner. Tumour core annotations based on the hVimentin staining were aligned and transferred to the corresponding γ-H2AX images by affine transformation to quantify γ-H2AX-positive cells within the tumour.

GICs for cilia validation were plated on laminin-coated coverslips and when they reached 60% confluency they were fixed with 4% PFA at room temperature for 15 min and subsequently washed with PBS. A permeabilization/blocking solution was prepared with 0.1% BSA, 1% Normal Donkey Serum, 0.1% Triton-X100 in PBS and it was added for 2 h at room temperature. Cells were stained for ace-α-Tubulin (Cell Signalling, #5335, used 1:800) and ARL13B (NeuroMab, 75-287, used 1:2000) overnight at 4 °C. After three washes with PBS and 1 h incubation with secondary antibodies diluted (1:250) in PBS, cells were washed again with PBS, counterstained with DAPI and mounted onto SUPERFROST slides (Thermoscientific, J1800AMNZ). Confocal microscope (Zeiss 880 Laser Scanning) was used to acquire images at 63x magnification using the Tiles tool. Each image contained 25 tiles (each at 63x) with an overlap of 10%. Images were analysed on ImageJ/FIJI using the cell counter tool. Manual count of cilia was conducted in blind conditions.

### Molecular biology

DNA and RNA extraction. For Formalin fixed paraffin embedded (FFPE) samples obtained either from primary and recurrent tumours (RNAseq, DNAme, and WES), or from murine PDX tumours (WES only) the tumour tissue was morphologically identified (through H&E staining) and marked on unstained slides. The highlighted regions were scraped off with a surgical scalpel and then collected. DNA and RNA were extracted from each of these tissue samples using the FFPE RNA/DNA Purification Plus Kit (Norgen, 54300) as per manufacturer protocol. For cells (GICs/GICRs/XGICs/XGICRs/IRGICs) both DNA and RNA were

extracted from pellets containing one well of a 6-well plate (confluency 80%) using the RNA/DNA/Protein Purification Plus Kit (Norgen, 47700) following the instructions provided by the manufacturer. The extracted DNA was further purified following the protocol of the Genomic DNA Clean & Concentrator-25 kit (Zymo research, D4011).

RT qPCR. RNA extracted from FFPE patient-tissue samples (see above) was retrotranscribed with SuperScriptIII (Invitrogen, 18080093). 5 ng of resulting cDNA were mixed with the right couple of primers (final concentration of 10 μM) and with Sybr Green PowerUp Master Mix (Applied Biosystems, A25742). The qPCR was performed on a StepOne Real-Time PCR System (ThermoFisher). The housekeeping gene used was GAPDH (F: CTGAGGCTCCCACCTTTCTC; R: TTATGGGAAAGCCAG TCCCC). Cilia markers used were C5orf49 (F: AATTTCCCACA GTTTAGCAG; R: ATACTTAGGAGTCACTCAGC), ARL13B (F: GAGCATGAGCAAATAGAGAC; R: TTAGCTGATTCCAATGA TGG), IFT88 (F: ATGTCTGCGTTTCTTAGTTC; R: CAACCTC TTCAGTTTTCTGG), RSPH11 (F: GGAAAGAGGAGAAGAGGA AG; R: AATTCAGTGATTTGGGTAGC).

### RNA sequencing

For library preparation, ribodepletion was carried out for FFPE samples and mRNA enrichment for cell samples, and 150 bp paired-end sequencing was then performed on a NovaSeq6000 platform. Reads were quality trimmed using trimgalore v0.6.5 and aligned to reference genome GRCh38.release109 using STAR v2.7.9a (Dobin et al, 2013). Reads were counted using feature-counts, and the ENSEMBL annotation Homo_sa-piens.GRCh38.109.gtf. Downstream analysis was performed in DESeq2 (Love et al, 2014), with filtering of genes with less than one read count per million (cpm) in all samples. For semi-supervised hierarchical clustering, the top 5000 variable genes (ranked by standard deviation) between GICs were used.

### DNA methylation

DNA methylation was assessed on the Illumina Infinium MethylationEpic v1 or v2 kits at the UCL Genomics, Institute of Child Health. EPIC microarray data were analysed using a cross-package Bioconductor workflow in R, employing the methylation-specific packages minfi, IlluminaHumanMethylationEPICanno.ilm10b4.hg19, missMethyl and DMRcate as well as limma as described in PMID:27347385. Briefly, raw data were imported in R using minfi and quality control was performed by assessing mean detection $p$-values of all CpGs in every sample to identify and remove potential failed samples. Normalisation was performed using minfi's preprocessQuantile method. Poor performing probes that failed in one or more samples were filtered out prior to differential methylation analysis retaining only good quality probes with detection $p$-value < 0.01, these were in total 835,820 probes. Probe-wise differential methylation analysis was performed using limma and probes with Benjamini and Hochberg adjusted $p$-value < 0.05 were considered statistically significant. Differential methylation analysis of regions was performed using the dmrcate function of R package DMRcate based on the annotated with genomics position M-values matrix and limma. For semi-supervised hierarchical clustering, the top 10,000 variable probes (ranked by standard deviation) between GICs were used.

### ScRNAseq

library preparation was performed at the Single Cell Genomics Facility (UCL, Cancer Institute) using the Chromium Next GEM Single Cell 3' Kit v3.1 (10X Genomics, 1000268) along with the Chromium Next GEM Chip G Single Cell Kit (10X Genomics, 1000120) according to the manufacturer's instructions. Pooled samples libraries were sequenced with lane sequencing on Novaseq 6000 S4 (Novogene). The Cell Ranger 2.0.1 pipeline was used to align reads to the GRCh38 human reference genome and produce count matrices for downstream preprocessing and analysis using the Seurat v4.0 R package (Hao et al, 2021). Samples, alignment was also performed against the Mm10 reference genome, and the number of aligning reads to each reference genome was calculated, with cells logFC _Hs_Mm < 1 excluded as contaminants. For quality filtration, thresholds of 500 genes and 1000 counts, and percentage mitochondrial thresholds of 20% were used. Expression values were library size corrected to 10,000 reads per cell and log1p transformed, with Principal component analysis (PCA) performed on the scaled data for the top 2000 variable genes. Batch correction was performed on principal components using Harmony (Kor-sunsky et al, 2019). Uniform Manifold Approximation Projection embeddings, Nearest Neighbours were calculated on the top 15 harmony-corrected PCA dimensions using Seurat's RunUMAP(), FindNeighbors() functions and clusters were assigned using the FindClusters() function with resolution = 0.25. Cluster marker genes were calculated using a Wilcoxon Rank Sum test, and differential expression analysis across experimental conditions was performed using MAST (Finak et al, 2015). For scRNAseq-based drug repurposing predictions, shared upregulated genes between primary vs modelled and primary vs true recurrence cells were ranked by their mean fold change, with the top 10 candidates for each patient queried against the drug gene interaction database v5 (Cannon et al, 2024). For reference signature scoring, average gene module expression was calculated for each single cell, subtracted by the aggregated expression of a random control set of features selected from the same average expression bins as the query genes (Venteicher et al, 2017). G/S and G2M gene modules are included in the Seurat v4.0 package; GSC cell state signatures were taken from Neftel et al (Neftel et al, 2019). For comparison to a reference dataset of 86 primary and recurrent patient tumours (GSE174554; (Wang et al, 2022)), we first used available cell metadata to remove tumour microenvironment cells, then restricted analyses to samples with at least 150 cells remaining, then scored cells for our recurrence-associated cilia (RAC) signature using Seurat's AddMo-duleScore() function.

## Statistical analysis

All statistical analysis and generation of graphs was performed using GraphPad Prism 9. Data are presented as mean ± SD. NS indicates non-significant with $P > 0.05$. $P < 0.05$ was considered statistically significant, with $p$ values $P \leq 0.05$, $P \leq 0.01$, $P \leq 0.001$ and $P \leq 0.0001$ represented with *, **, ***, ****, respectively. Differences between two groups were analysed by using the Student's unpaired t-test. Analysis of variance (ANOVA) was used to investigate more than two groups. Sample size varied according to the variability for each assay, for NGS experiments, wherever possible, we included two replicates (different passage numbers) for each condition. No data was excluded as outliers, and all

quantifications were performed in a blinded manner. Further information of the statistical analysis of specific datasets is indicated in the figure legends.

## Whole exome sequencing

Library preparation was performed using the xGen cfDNA & FFPE DNA Library Prep Kit (IDT), which includes addition of fixed single-strand UMIs during the first ligation step, which allows for the identification and correction of sequencing or PCR errors during analysis. Exome enrichment was carried out using Twist Human Comprehensive Exome kit and libraries were sequenced on a Novaseq6000 system to an average human genome coverage of 379x (range 81.8x–485.3x). Raw reads from each sequencing run and lane were separately trimmed for contaminating adaptor sequences using Skewer v.0.2.2 and then converted into unmapped bams with Picard's FastqToSam. UMIs were then extracted from the reads and inserted into read names (fgbio ExtractUmisFromBam). For non-PDX samples, samples were converted back to fastqs and aligned to the GRCh38 reference assembly of the human genome using the BWA-MEM algorithm v0.7.17. The mapped and unmapped bams were merged (with Picard's MergeBamAlignment) to generate a BAM with all necessary metadata and UMIs. UMIs could were then corrected (fgbio's CorrectUMIs) and collapsed into single read families via fgbio's GroupReadsByUmi and CallMolecularConsensusReads. Collapsed families were filtered using fgbio's FilterConsensusReads (--min-reads=2, only keeping collapsed families that had at least 2 supporting reads) and the final reads were clipped (fgbio's ClipBam) to prevent double counting of the same template which is important for SNV and copy number calling. For PDX samples, fastqs were instead aligned to a concatenated human-mouse (GRCh38/GRCm39) reference genome and analysed as above. After validating the final bam files, samtools was used to only keep read pairs that uniquely aligned to human chromosomes.

Somatic mutations were first called for each tumour sample separately against matched normal tissue with Mutect2 (v4.1.4.1) using the options '--af-of-alleles-not-in resource 0.0000025 --germline-resource af-onlygnomad.hg38.vcf.gz'. Variants detected in any tumour sample (marked PASS, coverage AD 10 in both normal (DNA extracted from syngeneic dura mater-derived fibroblasts) and tumour, at least 3 variant reads in the tumour, 0 variant reads in the normal, reference genotype in normal and non-reference genotype in tumour) were merged into a single list of candidate mutations. The multi-sample caller Platypus v0.8.1.1 was then used to recall variants at each candidate mutation position in all samples of the patient. In practice, this meant that the pipeline leveraged information across samples to improve the sensitivity of variant calling. The Platypus output of joint variant calls was then filtered to keep only high-quality variants with the flags PASS, alleleBias, QD or Q20, in canonical chromosomes (that is, not in decoy), a minimum number of reads NR > 10 and a genotyping quality GQ > 10 in all samples, a reference genotype (0/0) in the normal reference and a non-reference genotype (0/1 or 1/1) in at least one tumour sample. Finally (to account for any residual contaminating mouse reads), we calculated the highest VAFadj value (VAF = number of mutant reads/total number of reads, VAFadj = VAF/sample_purity) for each mutation across all samples of a patient. Variants were filtered out if the max VAFadj was less than 0.05.

### The paper explained

**Problem**

Glioblastoma is the most common and lethal primary brain tumour in adults. Despite multi-modal therapy (surgical excision, radiotherapy, and chemotherapy with temozolomide (TMZ)) all patients suffer recurrences seeded by glioma stem/initiating cells (GIC), and median survival remains only 15 months. Given the propensity of glioblastoma to recur, and the lack of clinical indications to re-operate at recurrence, a major goal should be to develop patient-specific predictive models of recurrence.

**Results**

Here we present an induced-recurrence PDX model (IR-PDX), in which mice xenografted with primary patient-derived GIC undergo treatment analogous to patient standard-of-care before tumours are allowed to recur. Using multi-modal sequencing and longitudinal patient-matched samples, we then compared induced-recurrence GIC to their true recurrence counterparts to benchmark the patient-specific accuracy of recurrence modelling.

**Impact**

We demonstrate how the IR-PDX model can yield biological insight into recurrence-associating phenotypes, highlighting how pharmacological ablation of cilia can resensitise recurrent GIC to TMZ. Further, we provide proof-of-concept evidence that genetic and transcriptomic analyses of IR-GIC can enable prospective precision medicine predictions that we then validated on true recurrent GIC. In the future such a recurrence model could be harnessed to speed-up the timeline to clinical intervention and prepare treatments for specific patients in advance, that could then be used upon their recurrences.

Maximum-parsimony trees were reconstructed with the Parsimony Ratchet method implemented in the phangorn R package v.2.8.1. If a sample contained any tumour variant reads for a mutation (provided that mutation had passed the above filtering) it was considered to be mutated (state 1) and others to be non-mutated (state 0). The acctran algorithm was used to estimate ancestral character states.

For Copy number calling, Sequenza-utils (version 3.0.0) was used to convert bams in 'seqz' files via 'bam2seqz' and 'seqz_binning' commands. The Sequenza R package could then be used compare each tumour sample against the matched normal fibroblast sample to assess (via changes in read depth and germline allele frequencies) the presence of copy number aberrations (CNAs). Sequenza also jointly estimates sample ploidy and purity. The CNAqc R package was used to combine mutations and CNAs and assess the purity and ploidy fits. Based on this analysis, Sequenza was rerun for the PDX samples with tweaked purity and ploidy parameters. CNAqc also smooths the copy number segments to get a cleaner copy number profile.

## Data availability

Data generated in this study have been deposited in public repositories and are available under the following accessing numbers. Gene expression omnibus (GEO); scRNAseq (GSE271619), DNAme (GSE271621), RNAseq (GSE271620). Sequence Reads Archive (SRA); WES (PRJNA1131147).

The source data of this paper are collected in the following database record: biostudies:S-SCDT-10_1038-S44321-025-00237-z.

## Peer review information

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

## Acknowledgements

This work is funded by grants from Brain Tumour Research (Centre of Excellence award to SM), Cancer Research UK (C23985/A29199 programme award to SM), Barts Charity (MGU0447 programme grant to SM). Part of the study was funded by the National Institute for Health Research to UCLH Biomedical research centre (BRC399/NS/RB/101410 to SB). SB was also supported by the Department of Health's NIHR Biomedical Research Centre's funding scheme. We thank Sara Badodi and Claire Vinel for sharing their knowledge and for support with in vivo experiments. We thank the staff of our animal facilities for help with the daily care of our mouse colony; Imran Uddin and Gordon Beattie, UCL Cancer Institute for library preparation and preliminary analysis of scRNAseq data; Julie Cleaver and Julie Foster, John Vane Science Centre Preclinical Imaging Core Facility and Maeve McLaughlin, Blizard Advanced Light Microscopy Facility for sharing their expertise. We acknowledge the use of data generated by the TCGA Research Network: https://www.cancer.gov/tcga.

## Author contributions

**Sara Lucchini**: Conceptualization; Data curation; Formal analysis; Validation; Investigation; Visualization; Methodology; Writing—original draft; Writing—review and editing. **James G Nicholson**: Conceptualization; Data curation; Formal analysis; Validation; Investigation; Visualization; Methodology; Writing—original draft; Writing—review and editing. **Xinyu Zhang**: Data curation; Investigation; Methodology. **Jacob Househam**: Data curation; Investigation; Methodology. **Yau Mun Lim**: Data curation; Investigation; Methodology. **Maximilian Mossner**: Data curation; Investigation; Methodology. **Thomas O Millner**: Data curation; Investigation; Methodology. **Sebastian Brandner**: Resources; Supervision; Funding acquisition. **Trevor Graham**: Resources; Supervision; Funding acquisition. **Silvia Marino**: Conceptualization; Resources; Supervision; Funding acquisition; Writing—original draft; Writing—review and editing.

Source data underlying figure panels in this paper may have individual authorship assigned. Where available, figure panel/source data authorship is listed in the following database record: biostudies:S-SCDT-10_1038-S44321-025-00237-z.

## Disclosure and competing interests statement

The authors declare no competing interests.

# Expanded View Figures

**Figure EV1. Histological characterization of PDX models.**

(A) Summary table of clinical and histo-molecular features of glioblastoma patients. (B) In vitro assessment of successful integration of F-Luciferase reporter genes in each GIC line used for this study. Left: quantification of Luminescence counts after imaging the cells with IVIS and measuring the ROIs ($n = 2$ for all lines; $n = 1$ for negative control). Error bars represent Mean ± SD. Right: 30 s kinetic measurement of firefly luminescence signal with plate reader ($n = 1$, with each line being measured with and without luciferin addition. Neg CTRL is represented by the average of all GIC without luciferin addition). (C) Curves showing the average tumour growth, measured via BLI, for all PDX derived from the injection of longitudinal primary and recurrent GICs: on the left both GBM39 and GBM39R ($n = 2$ and $n = 3$, respectively), while on the right GBM67 and GBM67R ($n = 3$). Error bars represent Mean ± SD. Each mouse's signal was normalized to the highest signal measured within the same animal over time. (D) Representative ICH staining for hNestin of each untreated PDX model generated in this study. Patient 39 both primary and recurrent tumours at the top and patient 67 at the bottom. Scale bars are 250 µm. (E) Representative H&E of both primary untreated PDX models generated in this study highlighting typica malignant features of glioblastoma (XGBM39 at the top and XGBM67 at the bottom). Black arrows indicate vascular proliferation; red arrows indicate mitosis. All images show spread nuclear atypia and high cellularity. Scale bars are 100 µm. (F) Representative ICH staining for KI67 (left) and cCaspase3 (right) of each untreated PDX model generated in this study. Patient 39 both primary and recurrent tumours at the top and patient 67 at the bottom. Scale bars are 100 µm.

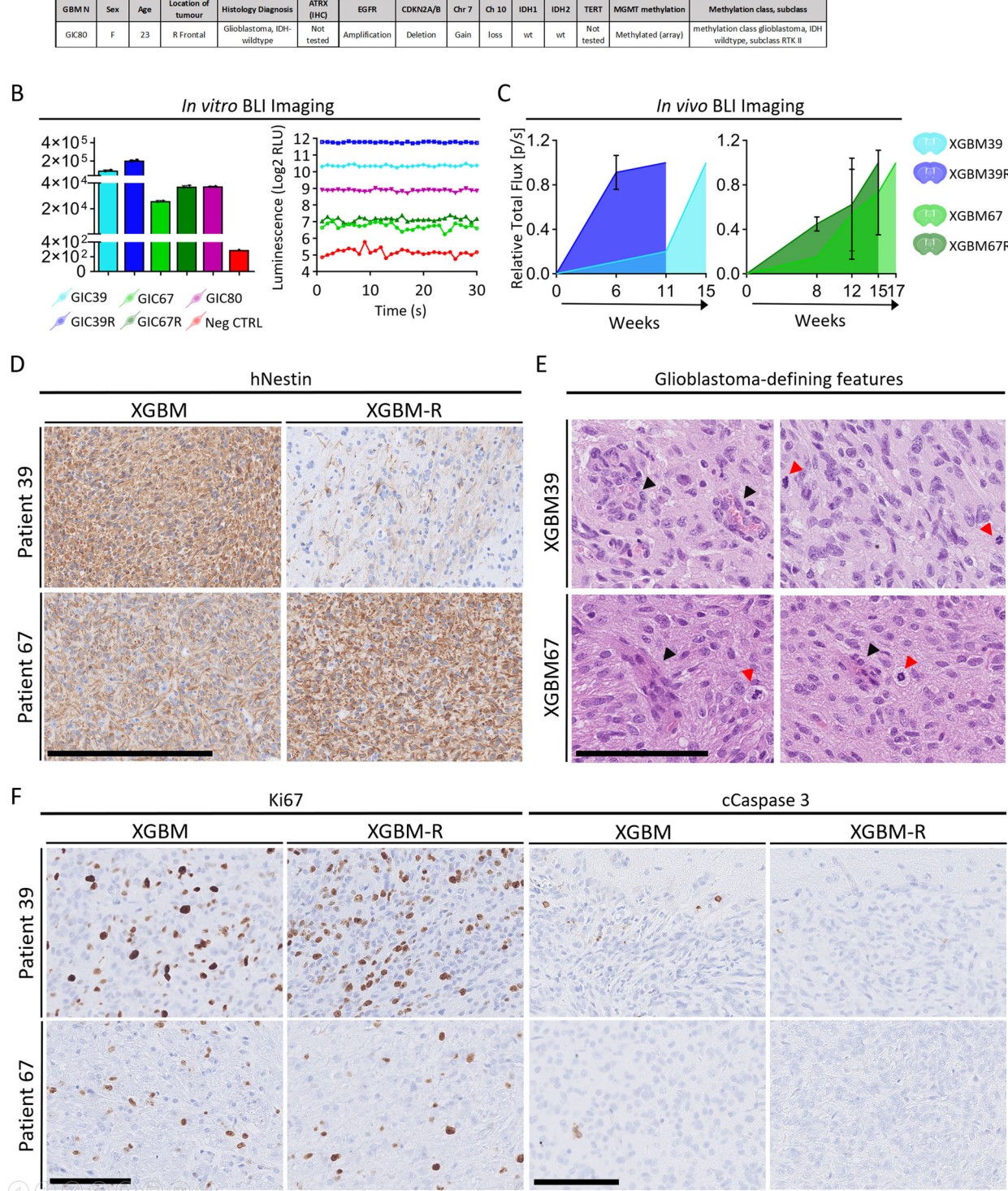

**A**

**Primary-Recurrent Pairs**

| GBM N | Sex | Age | Location of tumour | Histology Diagnosis | ATRX (IHC) | EGFR | CDKN2A/B | Chr 7 | Ch 10 | IDH1 | IDH2 | TERT | MGMT methylation | Methylation class, subclass | Treatment |
|---|---|---|---|---|---|---|---|---|---|---|---|---|---|---|---|
| GBM39 | M | 71 | Right Temporal | Glioblastoma, IDH-wildtype | Retained | No amplification | Deletion | Gain | loss | wt | wt | C228T | unmethylated (MSP PCR) | methylation class glioblastoma, IDH wildtype, subclass RTK I | Aug 2016: RT (60Gy/30#) + TMZ. Feb 2017: adj TMZ. June 2017 2nd surgery. |
| GBM39R | M | 72 | Temporal lobe, recurrence | Gliosarcoma, IDH-wildtype | Retained | Not tested | Not tested | Not tested | Not tested | wt | wt | Not tested | unmethylated (MSP PCR) | Not tested | |
| GBM67 | M | 51 | Right parietal | Glioblastoma, IDH-wildtype | Retained | Amplification | Deletion | Gain | loss | wt | wt | C228T | Unmethylated (Array) | methylation class glioblastoma, IDH wildtype, subclass RTK II | Mar 2017: RT (60Gy/30#) + TMZ. May 2017: adj TMZ. Feb 2018: 2nd surgery. |
| GBM67R | M | 52 | Right parietal | Recurrent glioblastoma, IDH-wildtype | Not tested | Not tested | Not tested | Not tested | Not tested | wt | wt | Not tested | unmethylated (MSP PCR) | Not tested | |

**MGMT methylated control**

| GBM N | Sex | Age | Location of tumour | Histology Diagnosis | ATRX (IHC) | EGFR | CDKN2A/B | Chr 7 | Ch 10 | IDH1 | IDH2 | TERT | MGMT methylation | Methylation class, subclass |
|---|---|---|---|---|---|---|---|---|---|---|---|---|---|---|
| GIC80 | F | 23 | R Frontal | Glioblastoma, IDH-wildtype | Not tested | Amplification | Deletion | Gain | loss | wt | wt | Not tested | Methylated (array) | methylation class glioblastoma, IDH wildtype, subclass RTK II |

**B** *In vitro* BLI Imaging

**C** *In vivo* BLI Imaging

GIC39  GIC67  GIC80
GIC39R  GIC67R  Neg CTRL

XGBM39
XGBM39R
XGBM67
XGBM67R

**D** hNestin

XGBM        XGBM-R

Patient 39

Patient 67

**E** Glioblastoma-defining features

XGBM39

XGBM67

**F** Ki67                                    cCaspase 3

XGBM        XGBM-R              XGBM        XGBM-R

Patient 39

Patient 67

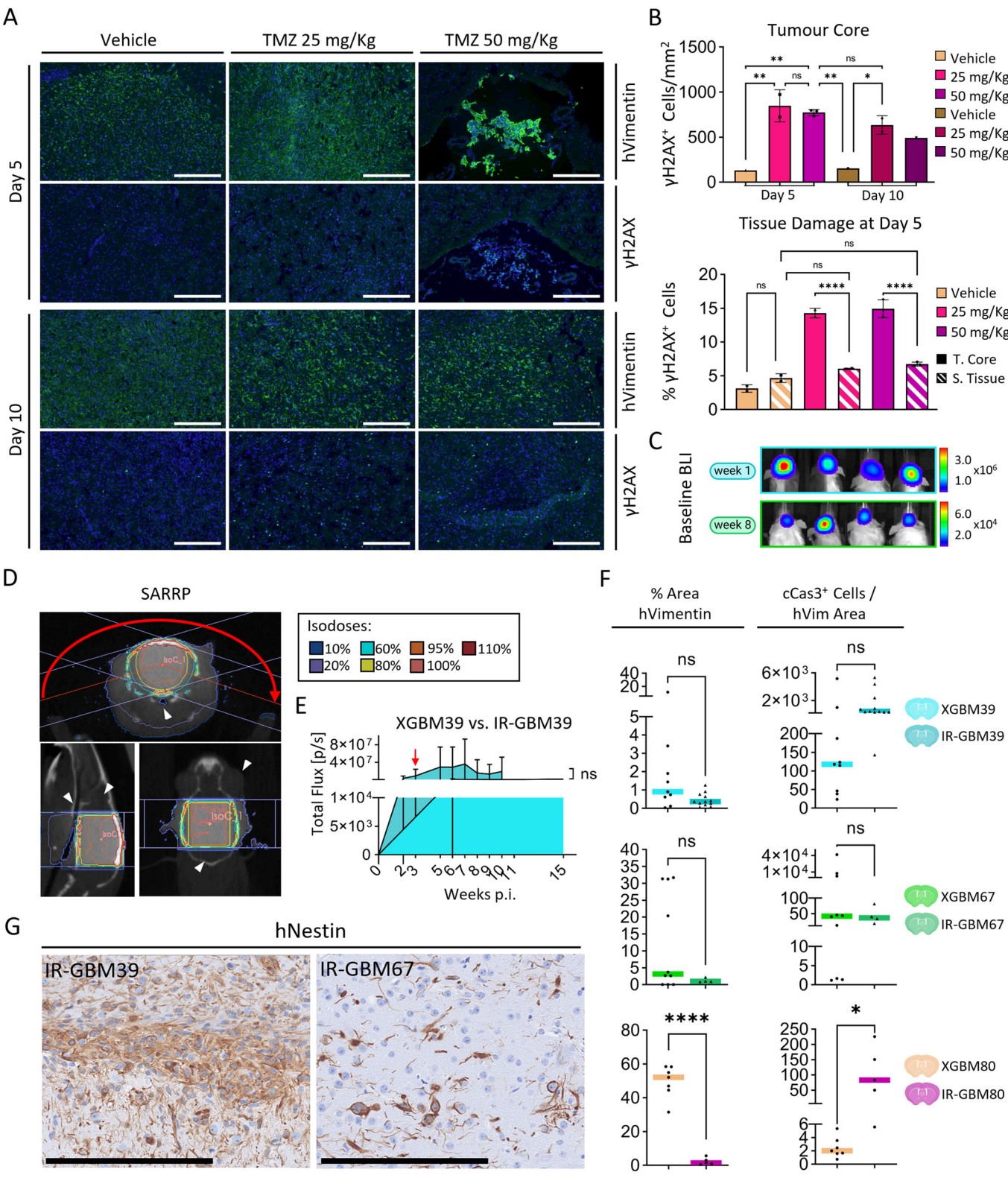

**Figure EV2.    Optimization of PDX model of induced recurrence.**

(A) TMZ tolerability assay: hVimentin and γH2AX IF stainings of brain sections for each time point and each administered dose. Scale bars are 250 μm. (B) Quantification of DNA damage at timepoints and doses of TMZ, tumour core and surrounding tissue are shown. Top: DNA damage quantification within the tumour core for each treatment group after 5 and 10 days of continuous administration. $n = 1$ for both vehicles, and Day10 50 mg/kg; $n = 2$ for both Day5 and Day10 25 mg/kg; $n = 3$ for Day5 50 mg/kg. Bottom: comparison of DNA damage between the tumour core and the surrounding tissue as an indication of the toxicity to non-neoplastic cells. $n = 2$ for both vehicles and both 25 mg/kg; $n = 3$ for both 50 mg/kg. One-way ANOVA. *$p = 0.0202$; [25 mg/kg vs. D5 Vehicle]: **$p = 0.0052$; [50 mg/kg vs. D5 Vehicle]: **$p = 0.0062$; [50 mg/kg vs. D10 Vehicle]: **$p = 0.0070$. ****$p < 0.0001$. Error bars represent Mean ± SD. (C) IVIS BLI of mice at the chosen timepoint of start of treatment. Each mouse showed a signal at 8 or 1 weeks post injection, respectively, for patients 39 (top) and 67 (bottom). (D) Radiation tolerability assay: CT image showing the isodoses of radiation received by the targeted tissue in one representative mouse. White arrows show successful avoidance of radiation-sensitive tissues. (E) Comparison of tumour growth curves, measured with BLI, between untreated primary PDXs and treated IR-PDXs derived by the injection of primary GIC39. The red arrow indicates a first scan performed after the sham injection and before radiation and TMZ administration. Error bars represent Mean ± SD. (F) Column scatter plots showing the quantification of immunohistochemical staining for hVimentin (%area of brain section surface which is positive for hVim; left) and cCaspase3 (right) in XGBM compared to IR-GBM. $n = 9$ for XGBM39, $n = 12$ for IR-GBM39, $n = 10$ for XGBM67, $n = 4$ for IR-GBM67, $n = 7$ for XGBM80, $n = 5$ for IR-GBM80. Unpaired t-test. ****$p < 0.0001$; *$p = 0.0106$. (G) Representative ICH staining for hNestin of both IR-PDX models generated in this study. Patient 39 on the left, patient 67 on the right. Scale bars are 250 μm. Source data are available online for this figure.

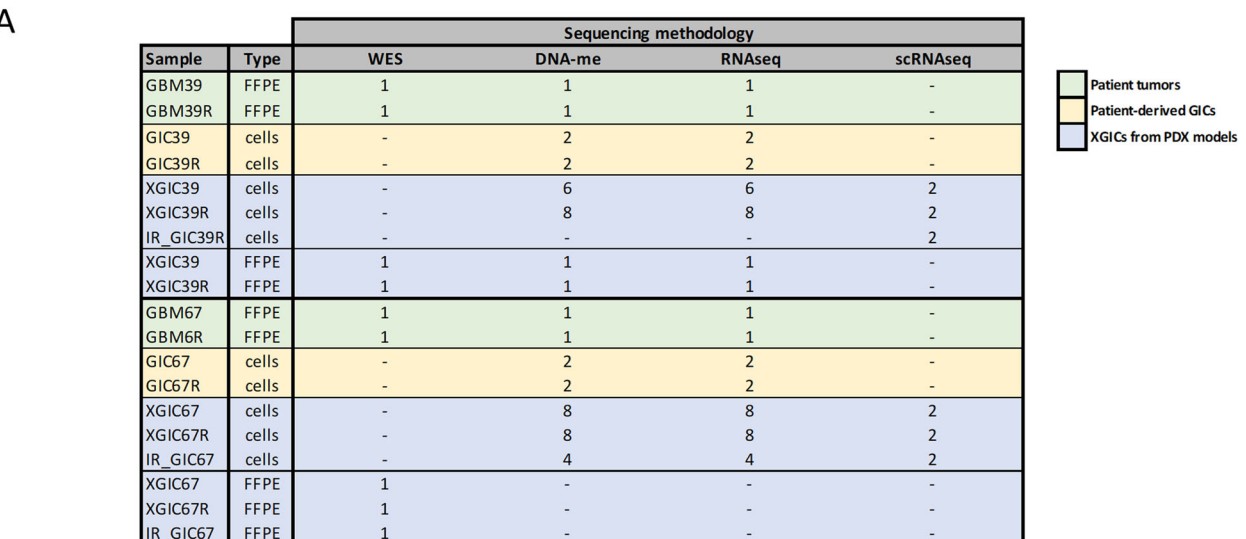

| | | Sequencing methodology | | | |
|---|---|---|---|---|---|
| Sample | Type | WES | DNA-me | RNAseq | scRNAseq |
| GBM39 | FFPE | 1 | 1 | 1 | - |
| GBM39R | FFPE | 1 | 1 | 1 | - |
| GIC39 | cells | - | 2 | 2 | - |
| GIC39R | cells | - | 2 | 2 | - |
| XGIC39 | cells | - | 6 | 6 | 2 |
| XGIC39R | cells | - | 8 | 8 | 2 |
| IR_GIC39R | cells | - | - | - | 2 |
| XGIC39 | FFPE | 1 | 1 | 1 | - |
| XGIC39R | FFPE | 1 | 1 | 1 | - |
| GBM67 | FFPE | 1 | 1 | 1 | - |
| GBM6R | FFPE | 1 | 1 | 1 | - |
| GIC67 | cells | - | 2 | 2 | - |
| GIC67R | cells | - | 2 | 2 | - |
| XGIC67 | cells | - | 8 | 8 | 2 |
| XGIC67R | cells | - | 8 | 8 | 2 |
| IR_GIC67 | cells | - | 4 | 4 | 2 |
| XGIC67 | FFPE | 1 | - | - | - |
| XGIC67R | FFPE | 1 | - | - | - |
| IR_GIC67 | FFPE | 1 | - | - | - |

Legend:
- Patient tumors
- Patient-derived GICs
- XGICs from PDX models

**Figure EV3. Whole exome sequencing of patient tumours and PDX models.**

(A) Summary of GBM samples used for multi-omic characterization of PDX models. (B) % of WES reads aligning to human vs mouse genome. (C) Total number of reads aligning to human vs mouse proportion of concatenated genome. (D) Total number mutations detected in WES samples. (E) Venn diagrams showing the number over overlapping mutations between primary, recurrent and modelled recurrence samples in patient 39 (left) and patient 67 (right).

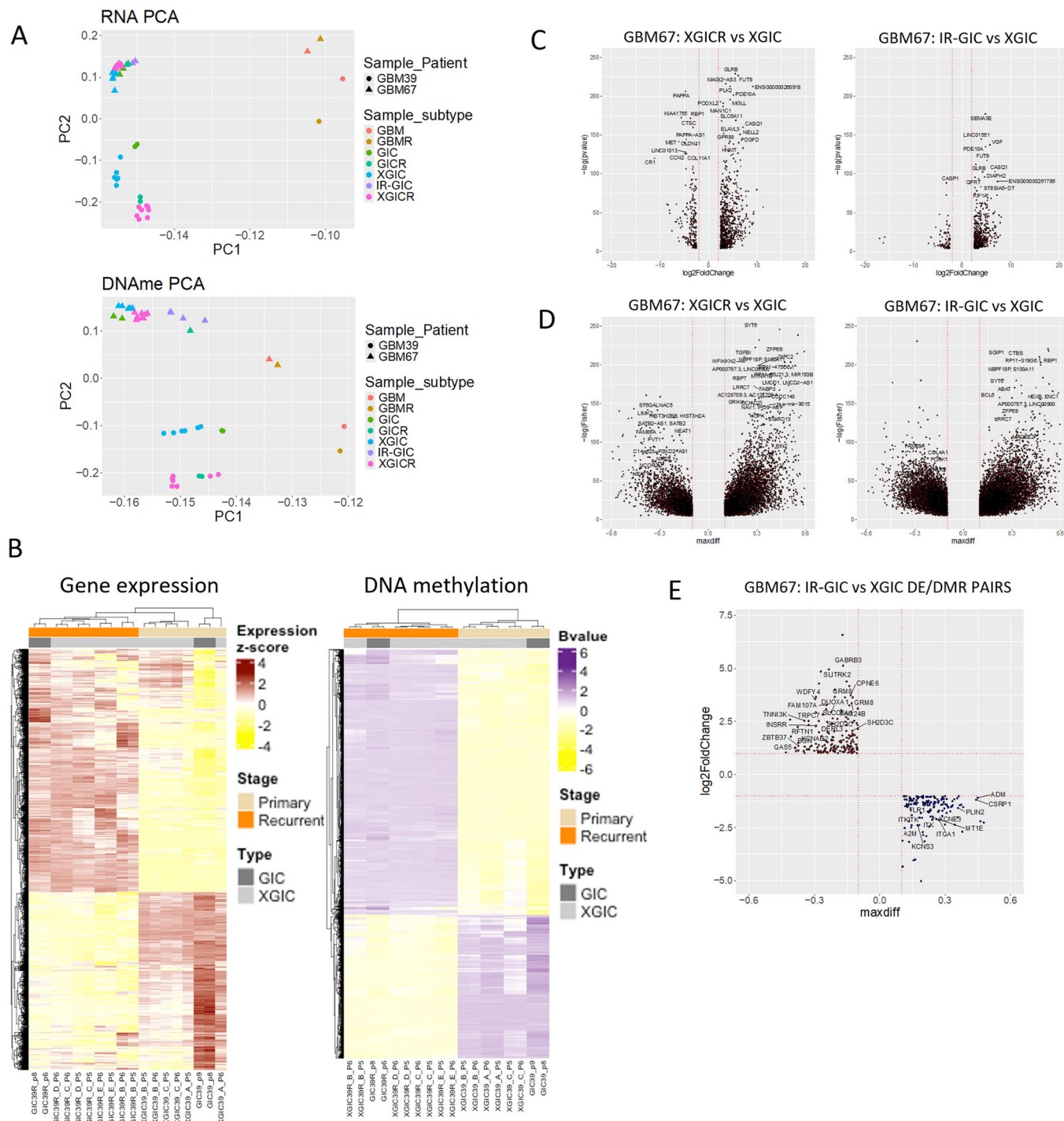

**Figure EV4. Transcriptional and epigenetic analysis of PDX models.**

(A) Principal component analysis of all RNA (top) and DNAme (bottom) GBM and patient- and xenograft-derived GIC samples. Samples are coloured according to their subtype (Primary GBM, recurrent GBM, patient-derived primary GICs or recurrent GICRs, xenograft-derived primary XGIC, induced-recurrence IR-GIC or true recurrence XGICR) and shaped according to their patient of origin with patient 39 shown as circles and patient 67 shown as triangles. (B) Semi-supervised hierarchical clustering of RNA (left) and DNAme (right) patient- and xenograft-derived GIC samples from patient 39. (C) Volcano plot of differential gene expression between XGIC67 and XGIC67R (left) or between XGIC67 and IR-GIC67 (right) samples. Y-axis shows −log(P-value) calculated in DEseq2 using a Wald's test. (D) Volcano plot of differential DNA methylation between XGIC67 and XGIC67R (left) or between XGIC67 and IR-GIC67 (right) samples. Y-axis shows -log(Fisher) calculated in DMRcate using a Fishers exact test. (E) Scatter plot of the concordantly regulated genes identified between GIC67 vs IR-GIC67 samples. Hypomethylated and overexpressed genes are red, hypermethylated and downregulated genes are blue. Labelled genes were also found in the XGIC vs XGICR comparison.

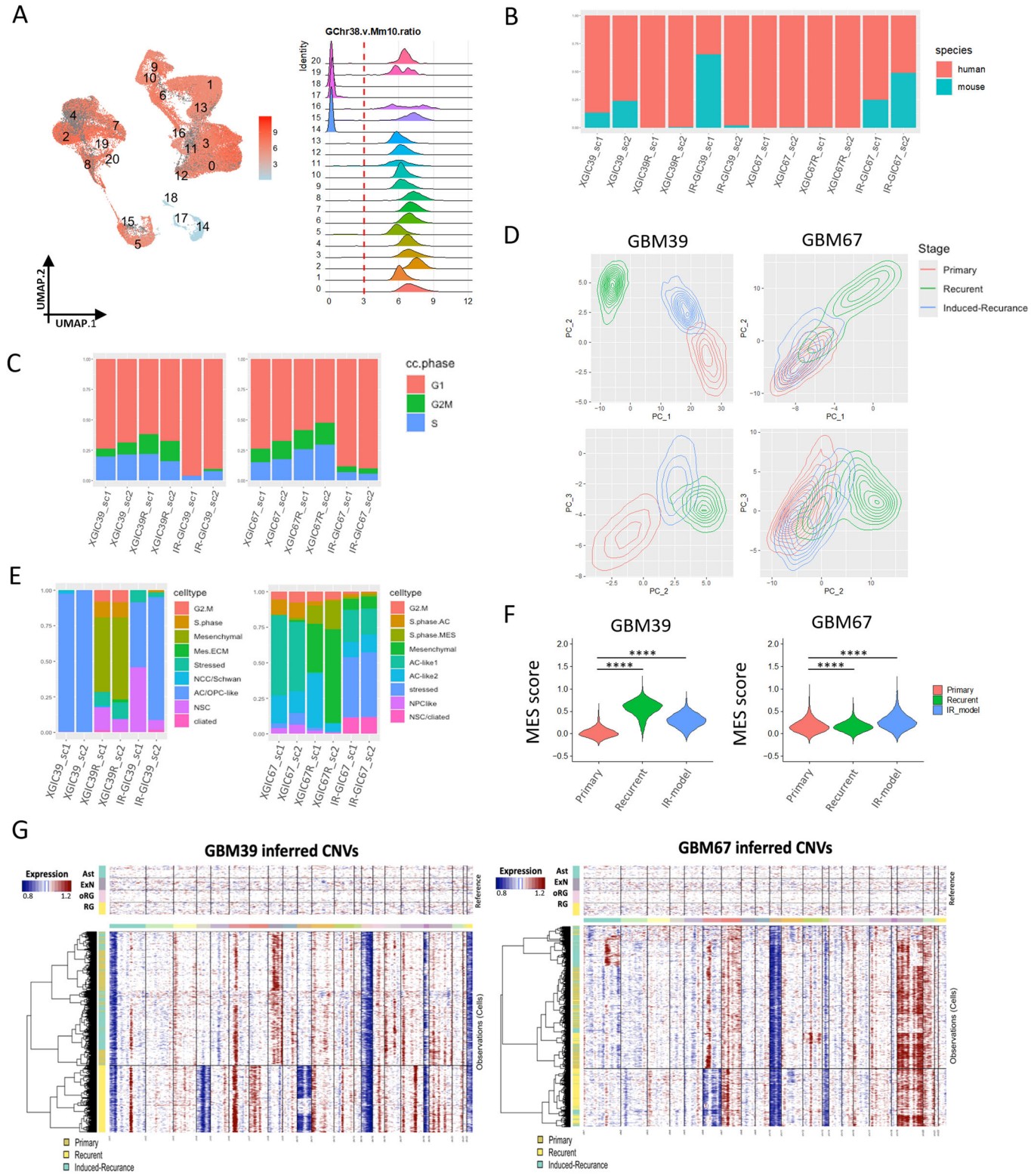

◀ **Figure EV5. scRNAseq of PDX models.**

(A) UMAP of PDX samples with cells coloured by ratio of reads aligning to human (GChr39) vs Mouse (Mm10) reference genome (left), ridge plot showing GChr39vsMm10 ratio cutoff used to remove clusters of Mouse cells (right). (B). Bar plot showing the proportion of human and mouse cells in each sample. (C) Bar plots showing the cell cycle phase proportion of cells in each sample of GBM39 (left) and GBM67 (right). (D) PCA plots of PDX cells. (E) Bar plots showing the cell type proportion in each sample of GBM39 (left) and GBM67 (right). (F) Violin plot showing Neftel et al MES-like signature scores in PDX samples. Wilcoxon test with Bonferroni correction (for GBM9 $n$ = Primary, 3727; Recurrent, 20638 IR-model, 979. For GBM67 $n$ = Primary, 20984; Recurrent 19099, IR-model, 2197). ****$p < 0.0001$. (G) Heatmap showing inferred Copy Number Variants in down samples PDX cells.

