## [Peer Review File · EMBO Molecular Medicine]

A novel model of glioblastoma recurrence to identify therapeutic vulnerabilities.

Sara Lucchini, James Nicholson, Xinyu Zhang, Jacob Househam, Yau Mun Lim, Maximilian Mossner, Thomas Millner, Sebastian Brandner, Trevor Graham, and Silvia Marino

Corresponding author: Silvia Marino (s.marino@qmul.ac.uk)

Review Timeline:

Submission Date:	17th Jul 24
Editorial Decision:	13th Aug 24
Revision Received:	19th Feb 25
Editorial Decision:	7th Mar 25
Revision Received:	21st Mar 25
Accepted:	25th Mar 25

Editor: Lise Roth

Transaction Report:

13th Aug 2024

Dear Prof. Marino,

Thank you for the submission of your manuscript to EMBO Molecular Medicine. We have now received the reports from two of the three referees who agreed to evaluate your manuscript. Unfortunately, referee #2 has not yet gotten back to us despite several chasers but given that both referees #1 and #3 provide similar recommendations, we prefer to make a decision now in order to avoid further delay in the process. Should referee #2 provide a report, we will send it to you, with the understanding that we will not ask you extensive experiments in addition to the ones required in the enclosed reports from referees #1 and #3.

As you will see below, both referees mention the novelty and translational interest of the work, however they also highlight several areas that should be improved, including (but not limited to) additional example(s) of targeted therapy, additional marker(s), and understanding of the functional relevance of cilia in recurrence.

Addressing these points and the other reviewers' concerns in full will be necessary for further considering the manuscript in our journal, and acceptance of the manuscript will entail a second round of review. EMBO Molecular Medicine encourages a single round of revision only and therefore, acceptance or rejection of the manuscript will depend on the completeness of your responses included in the next, final version of the manuscript. For this reason, and to save you from any frustrations in the end, I would strongly advise against returning an incomplete revision.

Based on the extent of the requested revisions, we are willing to extend the revision deadline to 6 months. If you would like to discuss further the points raised by the referees, I am available to do so via email or video. Let me know if you are interested in this option.

We require:

- 1) A .docx formatted version of the manuscript text (including legends for main figures, EV figures and tables). Please make sure that the changes are highlighted to be clearly visible.
- 2) Individual production quality figure files as .eps, .tif, .jpg (one file per figure). For guidance, download the 'Figure Guide PDF' (<https://www.embopress.org/page/journal/17574684/authorguide#figureformat>).
- 3) At EMBO Press we ask authors to provide source data for the main figures. Our source data coordinator will contact you to discuss which figure panels we would need source data for and will also provide you with helpful tips on how to upload and organize the files.
- 4) A .docx formatted letter INCLUDING the reviewers' reports and your detailed point-by-point responses to their comments. As part of the EMBO Press transparent editorial process, the point-by-point response is part of the Review Process File (RPF), which will be published alongside your paper.
- 5) A complete author checklist, which you can download from our author guidelines (<https://www.embopress.org/page/journal/17574684/authorguide#submissionofrevisions>). Please insert information in the checklist that is also reflected in the manuscript. The completed author checklist will also be part of the RPF.
- 6) All Materials and Methods need to be described in the main text using our 'Structured Methods' format, which is required for all research articles. According to this format, the Methods section includes a Reagents and Tools Table (listing key reagents, experimental models, software and relevant equipment and including their sources and relevant identifiers) followed by a Methods and Protocols section describing the methods using a step-by-step protocol format. The aim is to facilitate adoption of the methodologies across labs. More information on how to adhere to this format as well as a downloadable template (.docx) for the Reagents and Tools Table can be found in our author guidelines:
<https://www.embopress.org/page/journal/17574684/authorguide#structuredmethods>
- 7) Please note that all corresponding authors are required to supply an ORCID ID for their name upon submission of a revised manuscript.
- 8) It is mandatory to include a 'Data Availability' section after the Materials and Methods. Before submitting your revision, primary datasets produced in this study need to be deposited in an appropriate public database, and the accession numbers and database listed under 'Data Availability'. Please remember to provide a reviewer password if the datasets are not yet public (see

<https://www.embopress.org/page/journal/17574684/authorguide#dataavailability>).

9) For data quantification: please specify the name of the statistical test used to generate error bars and P values, the number (n) of independent experiments (specify technical or biological replicates) underlying each data point and the test used to calculate p-values in each figure legend. The figure legends should contain a basic description of n, P and the test applied. Graphs must include a description of the bars and the error bars (s.d., s.e.m.). Please provide exact p values.

10) Our journal encourages inclusion of *data citations in the reference list* to directly cite datasets that were re-used and obtained from public databases. Data citations in the article text are distinct from normal bibliographical citations and should directly link to the database records from which the data can be accessed. In the main text, data citations are formatted as follows: "Data ref: Smith et al, 2001" or "Data ref: NCBI Sequence Read Archive PRJNA342805, 2017". In the Reference list, data citations must be labeled with "[DATASET]". A data reference must provide the database name, accession number/identifiers and a resolvable link to the landing page from which the data can be accessed at the end of the reference. Further instructions are available at .

11) We replaced Supplementary Information with Expanded View (EV) Figures and Tables that are collapsible/expandable online. A maximum of 5 EV Figures can be typeset. EV Figures should be cited as 'Figure EV1, Figure EV2' etc... in the text and their respective legends should be included in the main text after the legends of regular figures.

12) The paper explained: EMBO Molecular Medicine articles are accompanied by a summary of the articles to emphasize the major findings in the paper and their medical implications for the non-specialist reader. Please provide a draft summary of your article highlighting

13) Author contributions: CRediT has replaced the traditional author contributions section because it offers a systematic machine readable author contributions format that allows for more effective research assessment. Please remove the Authors Contributions from the manuscript and use the free text boxes beneath each contributing author's name in our system to add specific details on the author's contribution. More information is available in our guide to authors.

16) As part of the EMBO Publications transparent editorial process initiative (see our Editorial at <http://embomolmed.embopress.org/content/2/9/329>), EMBO Molecular Medicine will publish online a Review Process File (RPF) to accompany accepted manuscripts.

In the event of acceptance, this file will be published in conjunction with your paper and will include the anonymous referee reports, your point-by-point response and all pertinent correspondence relating to the manuscript. Let us know whether you agree with the publication of the RPF and as here, if you want to remove or not any figures from it prior to publication. Please note that the Authors checklist will be published at the end of the RPF.

I look forward to receiving your revised manuscript.

Yours sincerely,

Lise Roth

***** Reviewer's comments *****

Referee #1 (Comments on Novelty/Model System for Author):

The generation of patient-derived glioblastoma recurrence models is of high importance to test new therapeutic strategies, that is why novelty and medical impact is high. Overall technical quality is high, only at few places statistical analysis was unclear

Referee #1 (Remarks for Author):

Lucchini et al. present a novel model of glioblastoma recurrence which is of high importance to the glioblastoma research community. The manuscript is well written, however several parts can be improved:

1. The authors show only one example of targeted treatment (Figure 3D) to verify the translational value of their model. This is not sufficient, the authors should show more examples.
2. All Patients should be characterised in the methods, IDH status, MGMT status, EGFR status, TERT promoter status, and Chromosome 7/10 status, and also briefly mentioned in the results. It is hard for the reader to extract this information. Only patient 80 is MGMT methylated?
3. Patient 80 (MGMT methylated) was not included in the genomic/transcriptomic/epigenomic analysis, why? It would strengthen the analysis to include MGMT methylated vs unmethylated samples as this is the only biomarker in GBM.
4. Figure 1D does not include any statistics, is there anything significant?
5. The authors only used human-specific vimentin. They should confirm with human-specific nestin as nestin is broadly expressed in GBM to see if it is retained in their models.
6. The Figure S2 is in the reviewers opinion more important compared to Figure 2 and parts may be moved into the main Figure. In Figure 2B and D no comparison to growth of primary untreated tumors is shown which would be important.
7. In Figure 4 B-F, the patient samples are not included why? It would be important to see in the PCA cluster where the patient samples are compared to GIC samples. The legends are too short. For an outside reader it is hard to grasp all the sample names.
8. Figure 5: What is the significance of cilia associated gene expression? Is it important for recurrence? The authors should knockdown central genes or if possible use drugs to show functional significance.

Referee #3 (Remarks for Author):

In this work, Lucchini et al describe a novel model to investigate glioblastoma (GBM) recurrence. They use primary and recurrent GBM cells from debulking surgery of 2 patients, and compare these with cells generated through their induced-recurrence (IR) model. These IR cells are generated by implanting patient primary GBM cells into brains of mice, treating with standard GBM therapy, and collecting cells that recur after treatment. Surgery of recurrent GBMs in human patients is rare, therefore the ability to compare true recurrence with induced recurrence is, to my knowledge, the novel aspect in this work and of great importance

in the field. Through exome sequencing, bulk RNA sequencing, DNA methylation analysis and single cell sequencing, the authors show that the IR-GBM cells replicate some of the changes that occur in a true recurrence. The authors conclude that the ability to perform IR prior to true recurrence gives important prognostic and treatment insights.

In general, the manuscript is well written and the experimental work is performed to a high standard. It would be useful if the following points were clarified:

N.B. there are no page or line numbers - making it difficult to explain where questions have arisen.

- 1) Results para1: Could the authors provide more information on how the GBM39 and GBM67 cell lines were derived? Were they derived for a previous study or by another group? Some information about the origin of these lines would be useful.
- 2) Were any analyses performed on cells following integration of the Firefly Luciferase reporter gene to ensure this integration did not affect key assay endpoints (e.g. proliferation, treatment resistance). Is it known whereabouts in the genome the reporter is stably introduced in each cell line?
- 3) "Real-time tumour growth was monitored with in vivo bioluminescence imaging (BLI), and upon the onset of symptoms,..." - could the authors provide more details on what symptoms they looked for as markers of GBM recurrence?
- 4) At the bottom of the first Results page, the authors state Fig1C shows histological analysis that confirms PDX tumours were "pleomorphic glial neoplasms with high mitotic activity, necrosis and vascular proliferation" - however Fig 1C shows hVimentin staining which I don't believe is sufficient information to show all those cellular properties. If so, could the authors outline how they did this?
- 5) The authors have included some exome sequencing metrics in the methods, but it would be useful to have some of these e.g. the depth of sequencing, in the results section so it's clear what data somatic mutations are being called on.
- 6) The authors state the results of differentially methylated gene analysis "shows the induced-recurrence model was able to capture aspects of recurrence-associated epigenetic regulation of gene expression." - while I agree with this statement to some degree, this isn't particularly quantifiable and doesn't give the true picture that the DNAm analysis doesn't really capture recurrence that well, compared to the differential gene expression analysis. I feel this should be said at this point in the results and discussed later.
- 7) Discussion - paragraph 1 "Through comprehensive multi-omic analyses, we showed that the IR-PDX model is capable of recapitulating genomic, epigenetic...". I feel this sentence is overselling the results shown as it implies that the IR model mimics true recurrence. I would say it is capable of recapitulating some genomic, epigenetic, etc. changes, but not all.
- 8) The authors discuss how the IR model can provide therapeutic insight prior to a true recurrence. The authors provide an example from their work through the identification and targeted treatment of a PIK3CA gain-of-function mutation in the GBM67R cell line. However its not stated if there were any other induced mutations that might be a therapeutic target or how well the induced recurrence mutations mimic'd the true recurrence variants? This may not be so straight forward without a true recurrence cell line. Would you treat a patient based on the IR findings alone?
- 9) Final paragraph: "... we demonstrate that our IR-PDX model can faithfully replicate patient-specific phenotypes associated with recurrence,..." - again I'm uneasy with the use of 'faithfully' in this sentence as this implies high level of concordance and reproducibility which isn't presented in this work. I agree that there is some replication, but to say this is faithfully replicating would require a much larger sample size than the n=2 presented here.
- 10) Methods "luminescence was recorded with a series of 5 sans..." - should this be scans?
- 11) Methods "TMZ was dissolved in DMSO DMSO" - duplicated DMSO.

Point-by-point rebuttal of individual reviewer comments:

Referee#1:

Lucchini et al. present a novel model of glioblastoma recurrence which is of high importance to the glioblastoma research community. The manuscript is well written, however several parts can be improved:

1. The authors show only one example of targeted treatment (Figure 3D) to verify the translational value of their model. This is not sufficient, the authors should show more examples.

To address this, we first queried whether further actionable mutations were detected in IR-GBM67 but none were found. We also newly sequenced IR-GBM39 bulk tumour (Rebuttal Figure 1) but, as expected given the lack of newly acquired mutations in GBM39R relative to the primary GBM39 tumour, we did not find actionable mutations in IR-GBM39 tumour either. We have chosen not to add these data to the revised manuscript because we feel the data are less robust than those previously obtained for GBM67. In particular, due to the small size of tumours in IR-GBM39 PDXs (Fig S1 G) it was technically challenging to scrape pure tumour tissue from FFPE sections for WES, with only ~10% (~20million) of IR-GBM39 reads aligning to the human genome (Fig R1 A-B), and a human exome coverage of 9.2X (compared to the previous minimum coverage of ~80X). Despite the lower coverage, a comparable number of total SNVs were detected (Fig R1 C) and major CNVs and cancer associated SNVs found in patient 39 were detected in the IR-GBM39 WES data (Fig R1 D-E). However, unlike for patient 67, we did not detect the presence of recurrence-specific mutations in the IR-GBM39, and maximum parsimony analysis did not place them as an intermediate between GBM39 and GBM39R. This may be due to patient specific differences in the clonal evolutionary response to our treatment regime – but due to the lower coverage of the IR-GBM39 sample we cannot exclude a technical artefact, hence the data not included in the revised manuscript.

Actionable mutations are found in only 7.2% of glioblastoma patients¹, it is therefore not unusual to fail to detect novel targets in modelled or true recurrences. However, we and others, have previously shown that GIC targeted therapeutics can also be predicted based on transcriptional data. We therefore mined transcriptional data for upregulated genes found both in IR-GICs and GICR and with known inhibitors in the drug gene interaction database (DGIdb v5²), choosing single cell data as it was available for both patients. Through this approach we identified Zonisamide a selective SCN4B sodium channel inhibitor for patient 67 as well as Pyrimethamine and TRAM34 for patient 39 which inhibits SLC47A2 (Multidrug And Toxin Extrusion Protein 2) and KCNN4 (Potassium Calcium-Activated Channel Subfamily N Member 4) respectively. In all cases the drug predictions were validated to be more effective in recurrent GIC relative to primary GIC (Fig 6E). These data strongly

suggest that when induced-recurrent modelling does not reveal genetic vulnerabilities, transcription-based drug predictions may be a viable alternative.

Rebuttal Figure 1 – Whole exome sequencing of patient tumors and PDX models

- % of WES reads aligning to human vs mouse genome
- Total number of reads aligning to human vs mouse proportion of concatenated genome
- Total number mutations detected in WES samples
- Genomic copy number analysis plots showing genomic fidelity of GIC PDX models to matched patient samples.
- Heatmaps clustering samples by Variant Allelic Frequency (VAF) at o pan-cancer driver mutations detected in GBM39 samples.
- Maximum-parsimony trees calculating relatedness between GBM39 WES samples.
- Venn diagrams showing the number over overlapping mutations between primary, recurrent and modelled recurrence samples in patient 39

2. All Patients should be characterised in the methods, IDH status, MGMT status, EGFR status, TERT promoter status, and Chromosome 7/10 status, and also briefly mentioned in the results. It is hard for the reader to extract this information. Only patient 80 is MGMT methylated?

We now include Supplementary table 1 summarizing the clinical and mutational status of all cell lines used in the manuscript and describe their characteristics in greater depth in the text. The reviewer is correct that only patient 80 has MGMT methylation, regrettably due to the scarcity of longitudinally paired GIC samples we were not able to include a MGMT-methylated pair in our analysis of the induced recurrence model. Instead, we used these cells to demonstrate that GIC responded to our induced-recurrence treatment regime in a similar fashion to patients, with the expected higher efficacy of treatment in the MGMT methylated line.

3. Patient 80 (MGMT methylated) was not included in the genomic, transcriptomic, or epigenomic analysis, why? It would strengthen the analysis to include MGMT methylated vs unmethylated samples as this is the only biomarker in GBM.

As described above, longitudinally paired MGMT methylated GIC were not available. We believe a key distinguishing feature of our study is the availability of the true recurrences we use to benchmark the efficacy of modelled recurrence. Without such cells available for GIC80 we did not feel it appropriate to include these cells in the (epi-)genomic analyses. Instead, these cells were used to demonstrate, that the GIC response to our induced-recurrence treatment regime is reflective of their MGMT methylation status.

4. Figure 1D does not include any statistics, is there anything significant?

Although some trends are apparent, there was relatively high inter-sample variability and none of the differences are significant between primary and recurrent PDXs. We have clearly stated this in the main text and added a note in the corresponding figure legend.

5. The authors only used human-specific vimentin. They should confirm with human-specific nestin as nestin is broadly expressed in GBM to see if it is retained in their models.

We have now included additional staining for human-NESTIN (XGBM and XGBMR samples in Fig S1D and IR-GBM samples in fig S2F), which yielded consistent results with human-VIM staining.

We note, that while the NES-staining of XGBM39R is clear and evident, it is comparably lower than XGBM39 and XGB67/R. Unfortunately, no tumour-core sections remained for this sample and the image presented represents the invasive front of the tumour, thus positive cells are sparser. Regardless, this image confirms NES expression is retained, and inspection of scRNAseq data showed high and equivalent NES expression across all sample conditions (Rebuttal figure 2).

Rebuttal Figure 2 - Nestin expression in scRNAseq data

6. The Figure S2 is in the reviewers opinion more important compared to Figure 2 and parts may be moved into the main Figure. In Figure 2B and D no comparison to growth of primary untreated tumors is shown which would be important.

We have taken the reviewers advice and moved the comparison of treatment response between unmethylated GIC67 PDX mice and PDXs generated with our methylated control cell line GIC80 to the main figure (now Fig 2E).

Comparison of bioluminescent growth curves between the treated induced-recurrence and untreated primary tumours is visualized in the new Fig 2E, and the additional panel Fig S2E. We have now also included quantification of tumour growth as measured by % Area hVimentin (Fig S2G) which shows a significant reduction in tumour size after treatment in PDXs generated from methylated GIC80, but not those generated in GIC39 or GIC67.

7. In Figure 4 B-F, the patient samples are not included why? It would be important to see in the PCA cluster where the patient samples are compared to GIC samples. The legends are too short. For an outside reader it is hard to grasp all the sample names.

We now add PCA plots including the primary and recurrent glioblastoma samples (Fig S4A) and have provided more details in the legends for Fig 4. The differences between glioblastoma and GIC samples are relatively large causing separation over the first principal component in both RNA and DNAm plots. This large difference partially obscures the differences seen between primary and recurrent GIC samples (Fig 4B). However, some recurrence-associated differences are still apparent, more so for the patient 39 samples, and particularly in the DNAm data (Fig S4A).

The difference between glioblastoma and GIC samples can be explained by their different sample processing (glioblastoma RNA/DNA was extracted from FFPE fixed sections and GIC RNA/DNA was extracted from frozen cells) as well as the fact that bulk glioblastoma samples contain not only GIC but also more differentiated tumour cells and stromal components. For this reason, we feel that

primary and recurrent GIC are the most appropriate control for benchmarking the efficacy of IR-GIC recurrence-modelling (Fig 4B-C).

Unfortunately, with only a single sample from primary and recurrent glioblastoma we do not have the statistical power to perform equivalent analyses for Fig 4D-F with the patient samples.

8. Figure 5: What is the significance of cilia associated gene expression? Is it important for recurrence? The authors should knockdown central genes or if possible use drugs to show functional significance.

To further explore the link between glioblastoma recurrence and ciliogenesis we pharmacologically ablated cilia used Mebendazole (MBZ⁵), building upon published studies in primary GIC ablating cilia (via siRNA against IFT88)³ or genetic increase in cilia (via shRNA against Nek2)⁴ which are now highlighted in the main text and discussed in greater depth.

We first confirmed that MBZ treatment leads to reduction in cilia in recurrent cells (Fig 6A). Next, we found that pharmacological ablation of cilia rendered recurrent GIC sensitive to TMZ again with significant synergy between MBZ and TMZ observed in both patients (Fig 6B-C). These new findings highlight the importance of cilia in mediating TMZ-resistance in recurrent glioblastoma and point towards potential therapeutic strategies for overcoming TMZ-resistance in recurrent patients.

Referee #3 (Remarks for Author):

In this work, Lucchini et al describe a novel model to investigate glioblastoma (GBM) recurrence. They use primary and recurrent GBM cells from debulking surgery of 2 patients, and compare these with cells generated through their induced-recurrence (IR) model. These IR cells are generated by implanting patient primary GBM cells into brains of mice, treating with standard GBM therapy, and collecting cells that recur after treatment. Surgery of recurrent GBMs in human patients is rare, therefore the ability to compare true recurrence with induced recurrence is, to my knowledge, the novel aspect in this work and of great importance in the field. Through exome sequencing, bulk RNA sequencing, DNA methylation analysis and single cell sequencing, the authors show that the IR-GBM cells replicate some of the changes that occur in a true recurrence. The authors conclude that the ability to perform IR prior to true recurrence gives important prognostic and treatment insights.

In general, the manuscript is well written, and the experimental work is performed to a high standard. It would be useful if the following points were clarified:

N.B. there are no page or line numbers - making it difficult to explain where questions have arisen.

Our apologies for this oversight – page and line numbers have been added.

1) Results para1: Could the authors provide more information on how the GBM39 and GBM67 cell lines were derived? Were they derived for a previous study or by another group? Some information about the origin of these lines would be useful.

GIC from patients 39 and 67 were derived in-house from bulk tumour surgical specimens as previously described⁶ in accordance with a published protocol⁷. Briefly, glioblastoma tissue was sliced and triturated with a razor blade, dissociated with Accumax (Sigma, A7089) at 37 °C for 10 min then filtered through a 70 µm cell strainer. Dissociated cells were plated on laminin-coated 6-well plate in NeuroCult NS-A Proliferation kit media (STEMCELL, 05751), heparin (2 µg/ml; Gibco 12587-010), mEGF (20 ng/ml, Prepro Tech, 315-09) and hFGF (10 ng/ml; Prepro tech, AF-100-18B). Established cells were passaged when 70% confluent, frozen in Stem Cell Banker (Ambio ZENOAQ, 11890), and stored in liquid nitrogen.

We have expanded our Materials and Methods section to better describe the derivation of these cells and made efforts to highlight their origin in the main text. Additionally, we now include Supplementary Table 1 which describes the molecular characteristics of the GIC used.

2) Were any analyses performed on cells following integration of the Firefly Luciferase reporter gene to ensure this integration did not affect key assay endpoints (e.g. proliferation, treatment resistance). Is it known whereabouts in the genome the reporter is stably introduced in each cell line?

We did not specifically investigate the effect of luciferase insertion on GIC biology, and indeed we do not know the exact whereabouts of the genomic insertion. However, we searched our whole exome sequencing data to confirm that the luciferase gene was not stably introduced within an exonic region in any of the cell lines sequenced. Briefly, the Luciferase construct sequence was appended to the human genome as a 'new chromosome' and reads were re-aligned. Searching aligned BAMs for reads confidently assigned to unique regions of the luciferase reference, with their paired read aligning to a region of the human exome (i.e. supporting an exonic insertion) did not yield results.

Whilst it is not impossible that intergenic or intronic luciferase insertion could impact key assay endpoints, we contend that random insertion would be unlikely to cause the same experimental artefacts twice. We therefore find it compelling that key findings, such as a recurrence-associated increase in cilia, and the rescuing of recurrent GICR TMZ-sensitivity upon inhibition of ciliogenesis is replicated across both patients.

3) "Real-time tumour growth was monitored with in vivo bioluminescence imaging (BLI), and upon the onset of symptoms,..." - could the authors provide more details on what symptoms they looked for as markers of GBM recurrence?

We have now listed the most typical neurological symptoms due to CNS pathology in the materials and methods. The evaluation of such symptoms for defining the end point of each mouse was performed in accordance with the scoring system laid out in our animal licence - held under the UK Animals (Home Office Guidelines: animals Scientific Procedures Act 1986, PPL 70/6452 and P78B6C064 Scientific Procedures) Act 1986. To our knowledge, there is no difference between neurological symptoms of primary and recurrent tumours.

We used BLI to confirm that treatment of the IR-GBM model resulted in a temporary reduction in tumour size, before resumption of tumour growth (Fig 2C). Based on this temporary efficacy of the treatment, we feel justified in our claims that neurological symptoms seen after this timepoint are those driven by recurrence. This claim is further supported by the various sequencing methods (WES, DNAm, RNAseq and scRNAseq) we use to demonstrate the molecular similarities between induced and true recurrence cells.

4) At the bottom of the first Results page, the authors state Fig1C shows histological analysis that confirms PDX tumours were "pleomorphic glial neoplasms with high mitotic activity, necrosis and vascular proliferation" - however Fig 1C shows hVimentin staining which I don't believe is sufficient information to show all those cellular properties. If so, could the authors outline how they did this?

We appreciate that the hVim and H&E presented in Fig 1C were not sufficiently magnified or annotated to clearly identify these features. We have now added 80X magnification H&E staining in Fig S1E where we have annotated mitotic activity, vascular proliferation and where nuclear atypia and high cellularity are more visible. Together with the new human-NESTIN staining (Figs S1D and S2F) we feel we have now robustly demonstrated the histological similarities between our PDX models and human glioblastoma.

5) The authors have included some exome sequencing metrics in the methods, but it would be useful to have some of these e.g. the depth of sequencing, in the results section so it's clear what data somatic mutations are being called on.

Whole exome sequencing samples were sequenced to an average human exome coverage of 379x (range 81.8x – 485.3x). We have now added the sequencing coverage to the main text to add context to the interpretation of somatic mutation results. As presented in Rebuttal figure 1 – we attempted to profile IR-GBM39 by WES during revisions, however this only yielded a human exome coverage of 9.2x, which we did not deem sufficiently high for inclusion in the final revised manuscript.

6) The authors state the results of differentially methylated gene analysis "shows the induced-recurrence model was able to capture aspects of recurrence-associated epigenetic regulation of gene expression." - while I agree with this statement to some degree, this isn't particularly quantifiable and doesn't give the true picture that the DNAm analysis doesn't really capture recurrence that well, compared to the differential gene expression analysis. I feel this should be said at this point in the results and discussed later.

The reviewer is correct to point out that the IR-GBM model captures more of the genes that are differentially expressed (~60%) than differentially methylated (~30%) upon recurrence. We have included an additional Venn diagram showing the shared differentially methylated genes (Fig 4C) and clearly referred to this figure in the main text. Additionally, we now reflect on this disparity in the discussion, and speculate that this may be due to the relatively slower rate of DNA methylation compared to transcriptional changes, and the accelerated recurrence in mice (weeks) relative to

human patients (months). Nevertheless, it is worth emphasising that the induced-recurrence model still identified more than 3000 genes that were also differentially methylated in the true recurrence.

7) Discussion - paragraph 1 "Through comprehensive multi-omic analyses, we showed that the IR-PDX model is capable of recapitulating genomic, epigenetic...". I feel this sentence is overselling the results shown as it implies that the IR model mimics true recurrence. I would say it is capable of recapitulating some genomic, epigenetic, etc. changes, but not all.

We acknowledge the reviewer's assertion that our IR model does not fully recapitulate/mimic true recurrence – indeed few, if any, disease models would rise to such a high bar. Instead, we sought to communicate that our IR model recapitulates some features of recurrence through multiple different methodologies and levels of comparison (i.e. genomic, epigenetic and transcriptional state changes). We have amended the sentence to both temper our claims and better clarify our intended point.

8) The authors discuss how the IR model can provide therapeutic insight prior to a true recurrence. The authors provide an example from their work through the identification and targeted treatment of a PIK3CA gain-of-function mutation in the GBM67R cell line. However, its not stated if there were any other induced mutations that might be a therapeutic target or how well the induced recurrence mutations mimic'd the true recurrence variants? This may not be so straight forward without a true recurrence cell line. Would you treat a patient based on the IR findings alone?

We searched for further actionable mutations in IR-GIC67 and (newly sequenced) IR-GIC39 cells (Rebuttal Fig 1) but none were found. There is a relative paucity of common targetable mutations in glioblastoma (seen in only 7.2% of patients¹), and hence we do feel it is not unusual to fail to detect novel targets in either modelled or true recurrences. In such situations, we now show that therapeutic predictions can also be derived from transcriptional data and validate three further drugs in Fig6 D-E.

Figure 3A only depicts mutations listed as COSMIC pan-cancer driver mutations, and does not capture the full extent of overlapping SNVs between samples which is now displayed in new Venn diagrams in FigS3E. We believe the higher overall overlap of the IR-PDX model with the recurrence relative to the primary and recurrent cells suggests our treatment regime is selecting for subclones which are also selected for during the natural recurrence process in patients.

The reviewer raises a very interesting point about the practicalities of future treatment of patients based on IR findings alone. Although we hope that ultimately this may be possible given it would tackle the lack of tissue/surgery at recurrence, embarking on such an undertaking would require a much larger cohort to provide data on the inter-patient reproducibility and accuracy of our approach. We would instead propose to start with a middle-ground approach whereby predictions are made based on findings from the IR-model and immediately tested for efficacy on GICR upon recurrence. If effective, they could then be used to treat patients post-operatively. This would bring forward the time to personalized clinical intervention by the weeks-months it would have taken to

sequence, analysed and make predictions based on the true recurrence. We now address these ideas and possibilities in the discussion.

9) *Final paragraph: "... we demonstrate that our IR-PDX model can faithfully replicate patient-specific phenotypes associated with recurrence,..." - again I'm uneasy with the use of 'faithfully' in this sentence as this implies high level of concordance and reproducibility which isn't presented in this work. I agree that there is some replication, but to say this is faithfully replicating would require a much larger sample size than the n=2 presented here.*

We feel that the levels of concordance shown between the IR model and recurrences (specific mutations, shared epigenetic and transcriptional state changes, cilia phenotypes) justify our use of *faithfully* in this final summary sentence. However, we do not claim that the IR model captures all features of recurrence and, at this reviewers direction, we have made efforts to make this clearer throughout our amended manuscript. We have added the word "some" to the final paragraph to help clarify this and temper our claims.

Further, we do not seek to make claims about the widespread reproducibility of our approach based on our limited cohort and have added additional discussion to this effect, when considering potential steps towards clinical translation of this work.

10) *Methods "luminescence was recorded with a series of 5 sans..." - should this be scans?*

This error has now been corrected

11) *Methods "TMZ was dissolved in DMSO DMSO" - duplicated DMSO.*

This error has now been corrected

Thank you in advance for considering this submission, which I now hope to be suitable for publication in *EMBO Molecular Medicine* and I am looking forward to hearing from you in due course.

Yours sincerely

Silvia Marino, MD FRCPath

Professor of Neuropathology, Queen Mary University of London
Lead, Barts Brain Tumour Centre
Director, Brain Tumour Research Centre of Excellence
Blizard Institute,
4 Newark Street, E1 2AT London, UK
Phone: 02078822585, Email: s.marino@qmul.ac.uk
<https://www.qmul.ac.uk/blizard/all-staff/profiles/silvia-marino.html>

References

- 1 Fougner, V. *et al.* Actionable alterations in glioblastoma: Insights from the implementation of genomic profiling as the standard of care from 2016 to 2023. *Neuro-Oncology Practice* (2024). <https://doi.org:10.1093/nop/npae082>
- 2 Cannon, M. *et al.* DGIdb 5.0: rebuilding the drug-gene interaction database for precision medicine and drug discovery platforms. *Nucleic Acids Res* **52**, D1227-D1235 (2024). <https://doi.org:10.1093/nar/gkad1040>
- 3 Wei, L. *et al.* Inhibition of Ciliogenesis Enhances the Cellular Sensitivity to Temozolomide and Ionizing Radiation in Human Glioblastoma Cells. *Biomed Environ Sci* **35**, 419-436 (2022). <https://doi.org:10.3967/bes2022.058>
- 4 Goranci-Buzhala, G. *et al.* Cilium induction triggers differentiation of glioma stem cells. *Cell Rep* **36**, 109656 (2021). <https://doi.org:10.1016/j.celrep.2021.109656>
- 5 Hong, J. *et al.* Mebendazole preferentially inhibits cilia formation and exerts anticancer activity by synergistically augmenting DNA damage. *Biomed Pharmacother* **174**, 116434 (2024). <https://doi.org:10.1016/j.biopha.2024.116434>
- 6 Vinel, C. *et al.* Comparative epigenetic analysis of tumour initiating cells and syngeneic EPSC-derived neural stem cells in glioblastoma. *Nat Commun* **12**, 6130 (2021). <https://doi.org:10.1038/s41467-021-26297-6>
- 7 Pollard, S. M. *et al.* Glioma stem cell lines expanded in adherent culture have tumor-specific phenotypes and are suitable for chemical and genetic screens. *Cell Stem Cell* **4**, 568-580 (2009). <https://doi.org:10.1016/j.stem.2009.03.014>

7th Mar 2025

Dear Prof. Marino,

Thank you for submitting your revised study. We have now received the reports from the referees. As you will see from the reports below, they are satisfied with the revisions, and I will therefore be able to accept your manuscript once the following editorial issues are addressed:

1/ Please address referee #1's remaining comment.

2/ Manuscript text:

- Please remove the yellow highlighted text, and only keep in track changes any new modification.
- Correct the order of the manuscript's sections: Abstract / Keywords / The Paper Explained / Introduction / Discussion / Methods / Data Availability / Acknowledgments / Disclosure and Competing Interests Statement / References / Figure Legends / Expanded View Figure Legends.
- Please provide up to 5 keywords.
- "Methods and Protocols" should be renamed "Methods".
- o Thank you for providing a reagent table. Please remove it from the manuscript text and upload as a separate file using our template.
- o Please include a statement that the experiments conformed to the principles set out in the WMA Declaration of Helsinki and the Department of Health and Human Services Belmont Report.
- o Please provide details on housing and husbandry conditions, and gender, age, and strain of the mice.
- o Statistics: please include a statement on sample size and exclusion/inclusion criteria.
- Data availability: thank you depositing your datasets. Please note that if practically possible and compatible with the individual consent agreement, we have to make sure that the authors deposit the human clinical datasets to public databases at the time of publication, however, authors must ensure that privacy of individuals is preserved.
- Acknowledgements: the funding information provided in this section should match the information provided in the submission system (should Barts Charity (MGU0447 programme grant to S.M.) and National Institute for Health Research to UCLH Biomedical research centre (BRC399/NS/RB/101410) be added to the list of funders in our system?)
- Author contributions: CRediT has replaced the traditional author contributions section because it offers a systematic machine readable author contributions format that allows for more effective research assessment. Please remove the Authors Contributions from the manuscript and use the free text boxes beneath each contributing author's name in our system to add specific details on the author's contribution. More information is available in our guide to authors.
- Please add a heading: "Disclosure and competing interests".
- Please correct the reference format to alphabetical order, with 10 author names listed before et. al.

3/ Figures:

- Please make sure that all figures are referenced in the text, in chronological order (currently, Fig. 2A is called out before Fig. 1B).
- Please carefully check the composition of Fig EV2, panel A, as similarities have been found between day 5, TMZ 25 mg/Kg and day 10, TMZ 50 mg/Kg. Please provide source data for this panel, and correct if needed.
- Please address the queries from our copy editors:
 1. Please define the annotated p values ****/***/**/* as well as provide the exact p-values for the same in the legend of figure EV2 B as appropriate.
 2. Please note that the exact p values are not provided in the legends of figures 5H, EV2 B, EV5 F.
 3. Please indicate the statistical test used for data analysis in the legends of figures 1B, 4E; EV2 B, D; EV4 C, D.
 4. Please note that the box plots need to be defined in terms of minima, maxima, centre, bounds of box and whiskers, and percentile in the legends of figures 4F
 5. Please note that information related to n is missing in the legends of figures 5F, EV1 B; EV2 B, D; EV5 F
 6. Please note that the error bars are not defined in the legends of figures 2C, E; 5H, EV1 B, C; EV2 B, D
 7. Please note that for heatmap present in figure EV5G a numbered scale bar is not provided. This needs to be rectified.

4/ Author checklist:

- please fill in the authors' info, top left corner
- please detail housing and husbandry conditions
- please provide information on inclusion/exclusion criteria (statistics)
- please fill in the subsections on ethics approval and informed consent/Helsinki (ethics)

5/ Synopsis:

I included minor edits in your text to fit our style and format, please let me know if you agree with the following or amend as you see fit:

"Unrepresentative pre-clinical models contribute to poor translation of glioblastoma research. We present an induced-recurrence xenograft model validated in longitudinal samples to uncover recurrence-associated phenotypes and patient-specific therapeutic vulnerabilities.

- A novel induced-recurrence (IR) model was established, in which mice xenografted with primary patient-derived glioma initiating/stem cells (GIC) are treated with a therapeutic regimen closely recapitulating patient standard-of-care.
- Comprehensive multi-omic analyses revealed that the IR model recapitulates aspects of genomic, epigenetic, and transcriptional state heterogeneity upon recurrence in a patient-specific manner.
- The IR model enabled both novel biological insights, including the positive association between glioblastoma recurrence and levels of Temozolomide-resistant ciliated tumour cells, and the identification of druggable patient-specific therapeutic vulnerabilities."

I have cropped a small portion of your synopsis (115x70 pixels) to serve as thumbnail on our website (attached). Please let me know if you agree, or provide a different image in the same dimensions.

6/ As part of the EMBO Publications transparent editorial process initiative (see our Editorial at <http://embomolmed.embopress.org/content/2/9/329>), EMBO Molecular Medicine will publish online a Review Process File (RPF) to accompany accepted manuscripts.

This file will be published in conjunction with your paper and will include the anonymous referee reports, your point-by-point response and all pertinent correspondence relating to the manuscript. Let us know whether you agree with the publication of the RPF and as here, if you want to remove or not any figures from it prior to publication.

I look forward to receiving your revised manuscript.

Yours sincerely,

Lise Roth

***** Reviewer's comments *****

Referee #1 (Comments on Novelty/Model System for Author):

Manuscript of high quality and relevance for the field.

Referee #1 (Remarks for Author):

The authors have comprehensively addressed all of my comments.

There is one error in Figure legend for Figure 6. Panels C and D describe the same, so one of them should be removed as this figure has panels A-D.

It would be great if the authors could next time label their figures, it makes it easier for referees.

Referee #3 (Remarks for Author):

I thank the authors for addressing the points laid out in my review and commend them for their work to update the manuscript. Having read the updated manuscript and point-by-point response, I feel my comments have been adequately addressed and have no further comments or queries.

The authors addressed the remaining editorial issues.

25th Mar 2025

Dear Prof. Marino,

Thank you for submitting your revised files. I am pleased to inform you that your manuscript is accepted for publication and is now being sent to our publisher to be included in the next available issue of EMBO Molecular Medicine!

With kind regards,

Lise Roth
